

# Moana Ocean Hindcast - a 25+ years simulation for New Zealand Waters using the ROMS v3.9 model.

Joao Marcos Azevedo Correia de Souza[1], Sutara H. Suanda[2,3], Phellipe P. Couto[1,3], Robert O. Smith[3], Colette Kerry[4], and Moninya Roughan[4]

[1]MetOcean Solutions, a division of Meteorological Service of New Zealand, Raglan 3225, New Zealand
[2]Department of Physics and Physical Oceanography, University of North Carolina Wilmington, USA
[3]Department of Marine Science, University of Otago, Otago 9016, New Zealand
[4]School of Biological Earth and Environmental Sciences, UNSW Sydney, NSW 2052, Australia

**Correspondence:** Joao M. A. C. Souza (j.souza@metocean.co.nz)

**Abstract.** Here we present the first open access long term 3D hydrodynamic ocean hindcast for the New Zealand ocean estate. The 28 year 5km x 5km resolution free running ocean model configuration was developed under the umbrella of the Moana Project, using the Regional Ocean Model System (ROMS) version 3.9. It includes an improved bathymetry, spectral tidal forcing at the boundaries, and inverse barometer effect usually absent from global simulations. The continuous integration provides a framework to improve our understanding of the ocean dynamics and connectivity, as well as identify long-term trends and drivers for particular processes. The simulation was compared to a series of satellite and *in-situ* observations, including sea surface temperature (SST), sea surface height (SSH), coastal water level and temperature stations, moored temperature time series, and temperature and salinity profiles from the CORA5.2 dataset - including Argo floats, XBT and CTD stations. These comparisons show the model simulation is consistent and represents important ocean processes at different temporal and spatial scales, from local to regional and from a few hours to years including extreme events. The root-mean-squared errors are 0.11m for SSH, 0.23$^{o}$C for SST, and <1$^{o}$C and 0.15g/kg for temperature and salinity profiles. Coastal tides are simulated well, and both high skill and correlation are found between modelled and observed sub-tidal sea level and water temperature stations. Moreover, cross-sections of the main currents around New Zealand show the simulation is consistent with transport, velocity structure, and variability reported in the available literature. This first multi-decadal, high resolution, open access hydrodynamic model represents a significant step forward for ocean sciences in the New Zealand region.

## 1 Introduction

Interest in the marine realm around New Zealand heavily centers on understanding the drivers of variability from event- to decadal time-scales in coastal areas. For ocean temperature, this is due to the sensitivity of valuable marine ecosystems to water temperature. While the New Zealand fishing industry expands to include coastal aquaculture efforts, there is increasing





evidence that global Western Boundary Current regions, including New Zealand, are rapidly warming (e.g., Shears and Bowen, 2017). For sea level, there is increasing interest in understanding and predicting elevation trends and how these interact with storm surge events and affect coastal infrastructure (e.g., Paulik et al., 2021). For ocean circulation, several efforts are underway to understand pollutant dispersal (e.g., Vennel et al., 2021) and biological (e.g., Chaput et al., 2022; Silva et al., 2019) connec-

tivity that impact water quality, species distribution, fishing and aquaculture activities. Although ocean observations provide an essential record of oceanic variability, they are intrinsically scarce - one simply cannot observe everything, everywhere, all the time. Therefore, a consistent, continuous, and realistic long integration of a regional simulation can serve as important source of information. A freely-available, multi-decadal, well-evaluated ocean state model that accurately and seamlessly represents coastal and open water variability across the entirety of New Zealand can provide many benefits to the Nation's maritime

industries and research purposes. This includes analysis of long-term trends, extreme events and planning for infrastructure developments, as well as research on the physical drivers of the ocean state.

The background ocean state from a free-running model is a key step towards predictive tools such as an ocean reanalyses that combines model and observations through a data assimilation scheme. A robust hindcast is necessary to avoid biases and represent the relevant physical processes for successful data assimilation. This becomes particularly important when imple-

menting strong-constraint assimilation schemes that assume the background model "perfectly" describes the system dynamics (Howes et al., 2017).

Regional simulations are also often used to provide boundary conditions to local models. Due the large differences between the spatial resolution needed for coastal and/or local studies (typically ≤1km) and the global simulations (typically ≈9km), intermediate or regional domains are necessary to transfer information from the large and meso-scales to the local domain.

As emphasized by Moore et al. (2019), surface and lateral boundary conditions can represent a significant source of error for such models. To maximize the benefits provided by regional model results, the simulation must include the main dynamical drivers (e.g., winds, tides, boundary currents, etc.), have a high enough spatial and temporal resolution to properly represent the regional processes of interest, and be evaluated against a variety of ocean observations to attest its realism.

Keeping these key concepts in mind, a 25+ years hindcast named the **"Moana Ocean Hindcast"** (Souza, 2022) was de-

veloped for the region around New Zealand. This simulation was performed under the umbrella of the Moana Project, that aims to revolutionize the understanding and prediction of ocean processes in New Zealand. A general project description is provided at **"www.moanaproject.org"**, and the model results are openly available (please visit the project website for details). The configuration developed for the hindcast and described in the present work was deployed operationally to provide 7 days forecasts daily, nested inside the Copernicus global ocean operational simulation.

A series of recent papers provide detailed descriptions of the main physical processes driving the circulation around New Zealand and its connection to the broader Pacific, Southern Ocean, and the global ocean circulation. Two publications in particular review the main ocean circulation features around New Zealand: Chiswell et al. (2015) describe the large scale currents and the "blue water" physical oceanography from the bibliography and recent satellite observations, hydrographic cruises, surface drifters and profiling floats; and Stevens et al. (2019) focus on the continental shelf waters and review prior

studies of ocean transport and mixing. In addition, Sutton and Bowen (2019) describes observed changes in ocean temperature





around New Zealand in the last 36 years, and identifies potential drivers of Marine Heat Wave events. The authors combine historical satellite "sea surface temperature" (SST) observations with water column temperature profiles to identify a warming trend in the waters North of the Subtropical Front that is highly correlated with air temperatures on interannual timescales. The strongest warming occurs in the southernmost limit of the local western boundary current, along the east coast of the North

Island south of East Cape. Salinger et al. (2020) go a step further and analyse the drivers of summer marine heatwaves. The authors conclude that the events were caused by either atmospheric fluxes or a combination of atmospheric forcing and ocean advection. More recently, Elzahaby et al. (2021) emphasizes the role of advection in driving deep and long-lasting marine heatwaves, highlighting the importance of properly representing the ocean currents for a good representation and predictability of such events. While sea surface temperature from satellites has been available for the last 30+ years, a high resolution integral

representation of the subsurface ocean structure can only be provided through a continuous model integration.

Given its dimensions and location in the South West Pacific basin, New Zealand regional circulation is subjected to significant influence from basin scale flow patterns introduced through the boundaries. Given their typical horizontal resolution, global reanalyses with DA are expected to represent such dynamics with a significant degree of skill. However, the complex bathymetry, narrow continental shelf, riverine influence, and the high mesoscale variability adds complexities that are beyond

the capability of relatively coarse global reanalysis, even considering comprehensive DA.

The repercussions for the regional dynamics are multiple. The interaction of dynamical processes normally excluded from the global simulations with the bathymetry significantly alters the water column structure. For example, this leads to a poor representation of the temperatures over the continental shelf due to miss-representation of coastal upwelling in areas such as the 3 kings Islands and the Bay of Plenty. The mass structure in the Firth of Thames can't be represented without the inclusion

of the input from the Waihou and Piako rivers, influencing the whole Hauraki Gulf. The circulation around the Pegasus and and Kaikoura canyons can't be reproduced without a high resolution bathymetry, affecting cross shore transports.

Here we document and describe the configuration of the 28 year free running hydrodynamic model and provide an evaluation of the simulation results so that the open access model can be used with confidence. The hindcast results and model configuration files are available at the Moana Project website (www.moanaproject.org). The operational forecast system, data

assimilating reanalysis and their ability in representing and predicting the ocean behaviour will be described in following publications. Section 2 presents the regional simulation configuration and the datasets used to force and validate the model; section 3 shows a rigorous model evaluation through comparisons between model results and a wide range of observations; and section 4 aggregates the conclusions on the overall quality and limitations of the Moana Ocean Hindcast model.

## 2 Methods

### 2.1 Model setup

In the present study, the Moana Ocean Hindcast regional model is developed using the Regional Ocean Modeling System (ROMS) version 3.9. This is a 3D primitive equations ocean model using hydrostatic and Boussinesq approximations. A full description of ROMS can be found in Shchepetkin and McWilliams (2005); McWilliams (2009) and the ROMS website



(www.myroms.org). ROMS has been particularly popular in the oceanographic community in the last decade. There are several

hindcasts and operational systems in place using this model, for regions as diverse as Hawaii (Matthews et al., 2012), the deep Gulf of Mexico (Maslo et al., 2020), and the East Australia Current (Kerry et al., 2016; Kerry and Roughan, 2020).

The Moana Ocean Hindcast regional model domain spans ≈161 to ≈185 $^o$E and from ≈52 to ≈31 $^o$S, with ≈**5km resolution** (Figure 1) with 467 x 397 grid cells. The grid limits were chosen to include the majority of the NZ EEZ, including the Auckland and Chatham islands, to place the open boundaries far enough from the 3 New Zealand main islands, and to have both the

Tasman Front to the north and the Subtropical Front to the south entering the domain through the west boundary. All this keeping in mind the computational cost for the provision of high resolution forecasts in an operational setting.



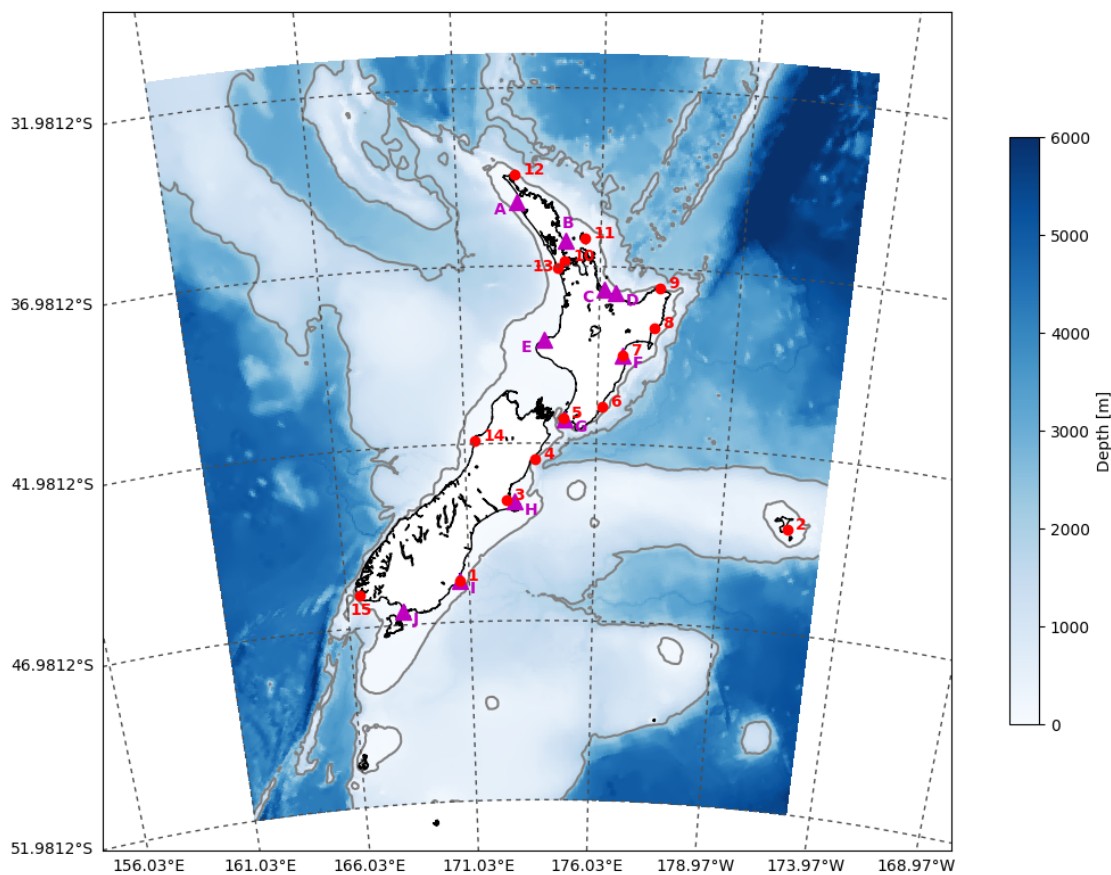

**Figure 1.** Moana Ocean Hindcast domain and bathymetry, and coastal observation locations of daily water temperature (magenta triangle) and tide-gauge sea level measurements (red dots) for use in model-data comparison (Section 3.4). Temperature locations are sequentially lettered (A - J) from North-to-South. LINZ tide gauge stations are numbered (1 - 15) to follow counter-clockwise Kelvin wave propagation from the southeast station (Port Chalmers) to the southwest (Purseygur). The gray contours show the positions of the the 200 m and 2000m isobaths.

ROMS uses a generalization of the classic terrain-following vertical coordinate system ($\sigma$-coordinates), defined as s-coordinates. Stretching functions are used to improve the resolution near the surface and bottom boundaries. In the present study we use the vertical stretching function proposed by Souza et al. (2015), that provides thinner and less variable surface layer. This is important for the inclusion of the assimilation of sea surface temperature, a following development step in the Moana Project.



**50 vertical layers** are used in the present configuration, with vertical stretching factors of 6 for the surface ($\theta_s$) and 2 for the bottom ($\theta_b$).

The model bathymetry was obtained from a combination of the "General Bathymetric Chart of the Oceans" (GEBCO) and
local sources. The smoothing of the bathymetry is commonly used in sigma coordinate models to avoid the generation of spurious pressure gradients (PGE - pressure gradient error) in regions of steep slopes due to the model discretization. An iterative approach was adopted to minimize this smoothing and avoid miss-representing the real basin geometry and, therefore, the dynamics. The smoothing was only applied to grid points where PGE associated bottom velocities were above the 1cm/s threshold, while preserving the total basin volume.

A split third-order upstream horizontal advection scheme (Marchesiello et al., 2009) is used for temperature and salinity to
help minimize spurious numerical diapycnal mixing in deep waters, while 4th-order centered differences scheme is used for the vertical advection. The vertical mixing was resolved using the "generic length scale" (GLS) turbulence model configured as a k-kl - equivalent to Mellor-Yamada 2.5 - as described by Warner et al. (2005). Along isopycnal horizontal mixing was defined for tracers, with along sigma levels mixing for momentum.

Atmospheric forcing was provided by the Climate Forecast System Reanalysis (CFSR) provided by National Center for At-
mospheric Research (NCAR) (https://climatedataguide.ucar.edu/climate-data/climate-forecast-system-reanalysis-cfsr). This includes 10m winds, air temperature, relative humidity, precipitation rate, downward short and long wave radiation, and sea level pressure. Upward long-wave radiation is calculated internally by ROMS. CFSR has been shown to work well in the NZ region (Souza et al., 2020). The fluxes from 42 rivers around New Zealand are included as climatological values obtained from the www.data.govt.nz portal. Following Janekovic and Powell (2012), tides were included at the open boundaries as a separate
spectral forcing with harmonics provided by the TPXO global tidal solution (Egbert and Erofeeva, 2002).

The configuration is nested inside daily results from the "Mercator Ocean Global Reanalysis" (GLORYS) 12v1 ocean reanalysis Lellouche et al. (2021) and this choice is described below. Radiation conditions were used for the tracers (temperature and salinity) in the open boundaries, associated to a nudging zone with time-scales decreasing from 1 day$^{-1}$ at the boundary to 0 at ≈200km towards the domain interior. The 3D velocities were clamped to the GLORYS fields. The free surface and
barotropic velocities used a combination of implicit Chapman and Flather boundary conditions, respectively. Solano et al. (2020) demonstrated these provide optimal results for the representation of tides in coastal models.

The Moana Ocean Hindcast model was run for 27 years, from January 1993 to December 2020. The first year was considered as a warm-up period, and discarded from the present analysis. The state variables, sea level, and velocity components were saved as hourly instantaneous fields and daily mean values. This provides an unprecedented source of high resolution
information, both spatial and temporal, on the ocean conditions and processes around New Zealand.

### 2.2 Boundary Conditions - Mercator reanalysis - GLORYS

A rigorous evaluation of the performance of 4 readily available global ocean reanalysis in the NZ region was conducted by Souza et al. (2020) who showed that the GLORYS12v1 performed best in the region when assessed against local observations. Although all 4 of the near global simulations analysed by Souza et al. (2020) (BRAN, HYCOM, GLORYS and CFSR) pre-





sented biases in the coastal region, GLORYS showed a more realistic ocean variability and smaller biases in the water column
structure in the offshore regions, making it suitable as boundary conditions.

The GLORYS ocean reanalysis is developed by the "Copernicus Marine Environment Monitoring Service" (CMEMS). It
has 1/12$^o$ horizontal resolution and 50 vertical levels. The reanalysis is generated using the "Nucleus for European Modelling
of the Ocean" (NEMO) ocean model driven at the surface by the ECMWF ERA-Interim reanalysis. It assimilates along track

altimeter observations (sea level anomaly), satellite sea surface temperature (SST), sea ice concentration and *in situ* temperature
and salinity vertical profiles from the "**C**oriolis **O**cean database **R**e**A**nalysis" (CORA) dataset (Szekely et al., 2019) using a
reduced-order Kalman Filter scheme. In addition, it uses a 3D-Var scheme for the correction of large-scale biases in temperature
and salinity. The reanalysis covers the satellite era from 1993 to 2018. For the years 2019 and 2020 the boundary conditions
are provided by nowcasts from Mercator Ocean operational model. This simulation uses the same model configuration as

GLORYS.

More details on GLORYS can be found in Lellouche et al. (2021) and the product page at the CMEMS website (http://marine.
copernicus.eu/services-portfolio/access-to-products/?option=com_csw&view=details&product_id=GLOBAL_REANALYSIS_
PHY_001_030).

## 2.3 Sea level variability forcing

Tides and the inverse barometer effect can be determinant for the representation of the the sea surface height and circulation
in coastal regions. These phenomena are usually not included in lower resolution global simulations that provide the boundary
conditions for regional models. At least in part, the poor performance of the global reanalyses in the NZ coastal region discussed
by Souza et al. (2020) can be explained by the absence of such key processes. To include tides, we obtained tidal constituents
from the Oregon State University TOPEX/Poseidon Global Inverse Solution (TPXO) version 7.8.1 (Egbert and Erofeeva,

2002). Following the methodology described by Janekovic and Powell (2012), 11 tidal constituents were introduced to our
simulation as spectral forcing at the boundaries to the free surface and barotropic velocity. The inverse barometer effect is
internally calculated in ROMS using the sea level pressure provided by CFSR.

## 2.4 Observational datasets for model evaluation

A number of publicly available observational datasets were chosen for model evaluation based on their spatial and temporal
coverage and the representation of the regional dynamics. The Moana Ocean Hindcast hindcast is validated against the obser-
vations and the GLORYS reanalysis to provide a comparison against the real world (obs) and an ocean state estimate used to
provide boundary conditions (GLORYS). By doing this we seek to highlight the improvement provided by the higher resolu-
tion (including bathymetry) and added physical forcing (tides, inverse barometer, rivers, etc) in the regional simulation. When
interpreting the results however, it is necessary to take into consideration that GLORYS assimilates the observations used for

the model evaluation while the Moana Ocean Hindcast is a free running simulation. It is then expected that GLORYS will
present smaller errors when compared to the assimilated observations associated with the large and meso-scale phenomena.





A selection of satellite derived products and vertical hydrographic profiles are used to evaluate how the simulation represents the large and meso-scale dynamics, while coastal sea level and long term temperature observations show the ability of the model in reproducing the hydrodynamic variability in shallow areas over the shelf.

### 2.4.1 Sea Surface Height (SSH) - CMEMS products

To evaluate the general pattern of the mean circulation, the simulations were compared to the Mean Sea Surface (MSS) topography and Sea Level Anomaly (SLA) satellite composite products provided by CMEMS (Pujol and Mertz, 2019). The MSS corresponds to a 20-year mean (1993-2012) based on altimetry data, provided at $1/60^o$ resolution. The SLA is provided as daily global maps on $1/4^o$ resolution. We use the CMEMS "all satellites" product which combines all the available along track observations at each time to provide the best possible estimate.

### 2.4.2 Sea Surface Temperature - NOAA OISSTv2.1

To evaluate the performance of the Moana Ocean Hindcast in reproducing Sea Surface Temperature (SST) we use the "Advanced Very High Resolution Radiometer Sea Surface Temperature" (AVHRSST) optimal interpolation SST product (OISST) provided by NOAA. OISST is an analysis constructed by combining observations from different platforms (satellites, ships, buoys and Argo floats) on a regular global grid. It consists of a $1/4^o$ horizontal resolution daily product, that covers the period from late 1981 to the present. More details on the SST product generation are provided by Reynolds et al. (2007).

### 2.4.3 Temperature and Salinity profiles - CORA 5.2 dataset

The CORA 5.2 dataset described by Szekely et al. (2019) provides a global comprehensive collection of *in situ* temperature and salinity profiles from 1950 to 2017. It contains data from a diverse set of observational platforms, from mechanical bathythermographs (MBT) prior to 1965; expendable bathythermographs (XBT); conductivity, temperature and pressure sensors (CTD), and Argo float profiles from the late 1990s forward. However, most of the observations are limited to 0-2000 m deep - the maximum depth of regular Argo floats. This dataset is used to evaluate the representation of the water column structure in the simulations.

### 2.4.4 Coastal water temperature and sea level

Sea level data are provided through a network of tide gauge stations maintained by Land Information New Zealand (LINZ) (Figure 1). Sea level data are collected at one minute sampling and available on-line since 2008 (https://www.linz.govt.nz/sea/tides/sea-level-data/sea-level-data-downloads). Fifteen locations in the tide gauge network are used for model-data evaluation of tidal and sub-tidal sea level variability (Table 1).




**Table 1.** Land Information New Zealand (LINZ) tide gauge station names, ID and locations for model-data evaluation of sea level.

| Tide gauge station | LINZ ID | Latitude ($^o$N) | Longitude ($^o$E) |
|---|---|---|---|
| 1. Port Chalmers | OTAT | -45.82 | 170.65 |
| 2. Chatam Islands | CHIT | -44.03 | 183.63 |
| 3. Sumner | SUMN | -43.57 | 172.57 |
| 4. Kaikōura | KAIT | -42.42 | 173.70 |
| 5. Wellington | WLGT | -41.28 | 174.78 |
| 6. Castle Point | CPIT | -40.92 | 176.22 |
| 7. Napier | NAPT | -39.48 | 176.92 |
| 8. Gisbourne | GIST | -38.67 | 178.03 |
| 9. Lottin Point | LOTT | -37.55 | 178.17 |
| 10. Auckland | AUCT | -36.83 | 174.78 |
| 11. G. Barrier Island | GBIT | -36.18 | 175.48 |
| 12. North Cape | NCPT | -34.42 | 173.05 |
| 13. Manukau | MNKT | -37.05 | 174.52 |
| 14. Charleston | CHST | -41.90 | 171.43 |
| 15. Puysegur | PUYT | -46.08 | 166.58 |

Ten locations around New Zealand collect daily water temperature measurements from shore (Table 2). Seven stations are
collected by the New Zealand's "National Institute of Water and Atmospheric Research" (NIWA) with digital temperature
sensors (Chiswell and Grant, 2019). Additional data are obtained from the University of Otago Portobello Marine Labora-
tory (daily measurement with hand-held mercury thermometer), the University of Auckland Leigh Marine Laboratory, and
a Datawell Waverider buoy maintained by the Port of Tauranga. Data record continuity and duration varies considerably
throughout the hindcast period, with some stations reporting near complete coverage over the 27 year period (e.g., Evans
Bay, Portobello), while other datasets extend ≈ 4 years (e.g., Bluff). Efforts continue to centrally collate and archive oceanic
measurements on the New Zealand Ocean Data Network (NZODN,https://nzodn.nz/portal/) and to build co-ordinated ocean
observing partnerships across the nation NZ-OOS (e.g., Callaghan et al., 2019).




**Table 2.** Coastal daily surface temperature measurement station names, locations and data coverage, where percentage represents the duration of the Moana Ocean Hindcast period. Measurement stations are maintained by NIWA, except where noted, *Leigh Marine Laboratory, University of Auckland, **Tauranga wave buoy, Bay of Plenty Regional Council, ***Portobello Marine Laboratory, University of Otago.

| SST station | Latitude ($^o$N) | Longitude ($^o$E) | Length of time series (yrs) | % Data coverage |
|---|---|---|---|---|
| A. Ahipara | -35.17 | 173.10 | 14.84 | 98% |
| B. Leigh* | -36.27 | 174.80 | 17.34 | 98% |
| C. Moturiki | -37.63 | 176.18 | 15.28 | 100% |
| D. Tauranga** | -37.70 | 176.62 | 14.27 | 78% |
| E. New Plymouth | -39.05 | 174.03 | 10.85 | 87% |
| F. Napier | -39.48 | 176.92 | 11.60 | 88% |
| G. Evans Bay | -41.30 | 174.80 | 24 | 93% |
| H. Lytleton | -43.63 | 172.90 | 14.61 | 99% |
| I. Portobello*** | -45.83 | 170.65 | 24 | 98% |
| J. Bluff | -46.60 | 168.30 | 3.58 | 100% |

## 3 Results and discussion

### 3.1 Surface

Daily mean fields for SSH and SST were calculated from the Moana Ocean Hindcast to make it comparable to GLORYS reanalysis and the AVISO and OISST observational products.

The Moana Ocean Hindcast reproduces well both the large and meso-scale SSH structure (Figure 2). The Moana Ocean Hindcast temporal mean SSH agrees with the GLORYS reanalysis, that assimilates altimeter observations. It shows the main high and low SSH centres and their respective fronts that reflect the positions of the main large-scale currents at the same

locations (Figure 2). Gradients in SSH are generally stronger in the Moana Ocean Hindcast, specially in the region of the East Auckland Current (EAUC) to the north and east of the North Island of New Zealand. This constitutes in sharper fronts and stronger boundary currents, desired in a higher resolution model as discussed in section 3.3.

Figure 3 shows a similar pattern for the variance of the SSH between the Moana Ocean Hindcast and GLORYS simulations, and the gridded AVISO observational product. The Moana Ocean Hindcast shows larger overall variance, as expected due to

its higher horizontal resolution. These larger values are more evident in the regions corresponding to the eddying EAUC and its continuation to the east in the Subtropical Front, and the area influenced by the northern branch of the Antarctic Circumpolar Current in the southeast extremity of the domain. Ballarotta et al. (2019) showed that the effective spatial resolution of the altimetry maps around New Zealand is between 150 and 200 km, marginally resolving the mesoscale eddies.

To evaluate the spatio-temporal structure of the SSH variability the elevation fields from the Moana Ocean Hindcast and

GLORYS simulations and the AVISO observations were decomposed into the empirical orthogonal functions (EOF). Following



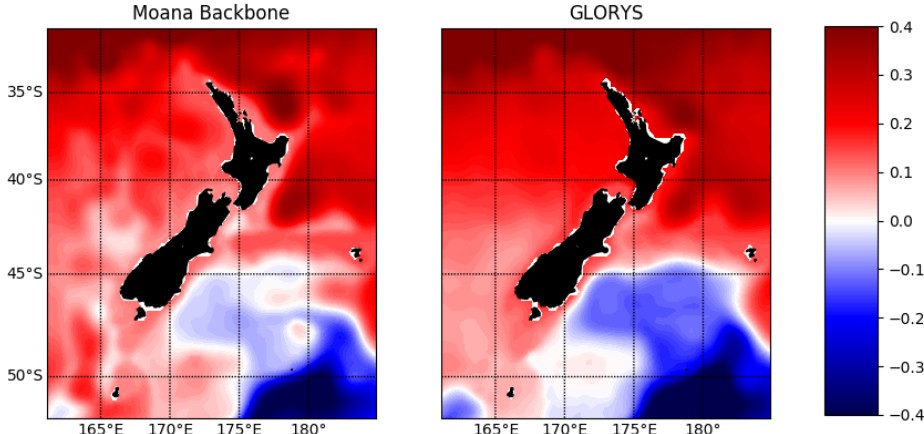

**Figure 2.** Temporal mean SSH (m) from the free-running Moana Ocean Hindcast (left) and the data-assimilating GLORYS (right) simulations. The general patterns of the SSH are reproduced by both models.

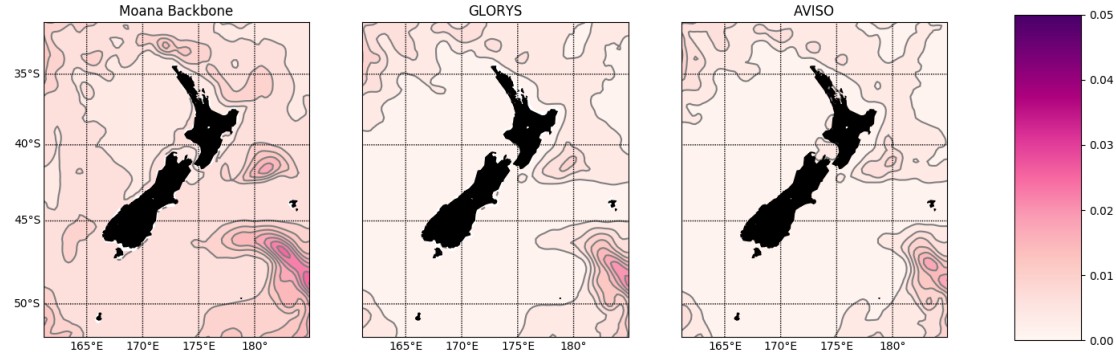

**Figure 3.** Variance of the SSH ($m^2$) from the free-sunning Moana Ocean Hindcast (left) the data-assimilating GLORYS (centre) simulations, and the AVISO product (right). Although the same general distribution is observed, the Moana Ocean Hindcast presents stronger variability due to its higher spatial resolution.





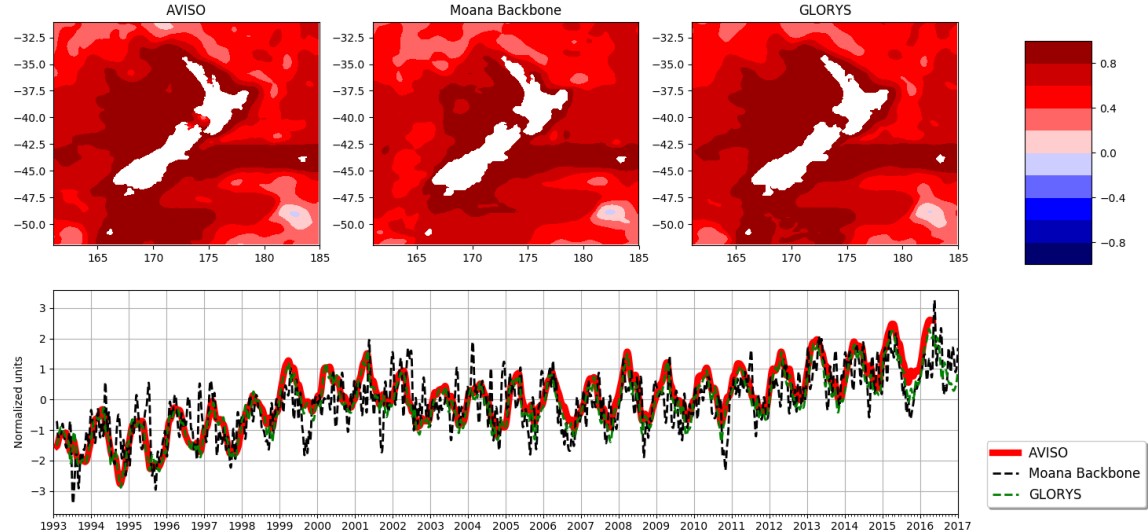

**Figure 4.** Empirical orthogonal function (top maps) and principal component (bottom time-series) decomposition of the 40 days low-pass filtered SSH from the Moana Ocean Hindcast and GlORYS simulations, and the AVISO observational product.

Ballarotta et al. (2019), a **40 days** low-pass filter was used on the data prior to the EOF decomposition. The authors show that the mean temporal resolution of the AVISO altimetry maps in the equator is  34 days, with values ranging between 35 and 42 days in New Zealand. Figure 4 shows the first EOF, that explains 35% / 38% / 37% of the variance for the Moana Ocean Hindcast / GLORYS / AVISO, and the remainder EOFs having contributions one order of magnitude smaller. The overall
pattern is the similar between the simulations and observations. The Moana Ocean Hindcast shows stronger high-frequency variability, as evidenced in the principal-component time series. This can be related to a series of factors, including the higher horizontal resolution and the inclusion of physical processes such as tides and the inverse barometer effect. The seasonal to inter annual variability of the SSH throughout the domain is well reproduced.

The Moana Ocean Hindcast also reproduces well the SST throughout the domain, with good representation of variability in
a range of time scales from sub-seasonal to inter-annual as shown in Figure 5. It is interesting to observe how the historical high temperature peak in 2018 is well reproduced in the simulation. Individual high temperature anomaly events with duration of the order of a few days, such as marine heatwaves, were also reproduced and will be explored in depth in separate publications.

The RMSE and BIAS maps in Fig 6 show deviations of the model in relation to the OISST observational products. The model errors are concentrated in the coastal waters and the position of strong eddying fronts. The BIAS pattern is reminiscent
of the differences showed by GLORYS in Figure 7 of Souza et al. (2020). These differences relate to a series of factors: (a) The fact that a free running simulation is in general not able to place eddies in the exact same place and time of the observations, (b) the relatively coarse resolution of the OISST product that tends to smooth the frontal regions, and (c) issues related to the





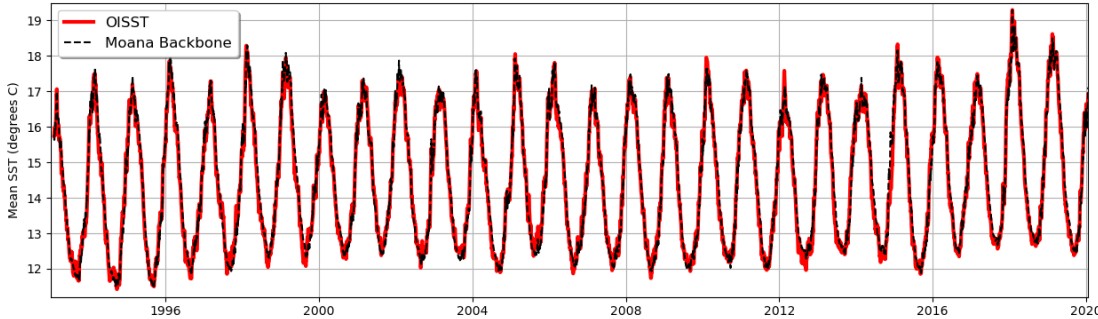

**Figure 5.** Domain averaged SST ($^{o}$C) for the Moana Ocean Hindcast and the OISST observational product. The Moana Ocean Hindcast model is able to reproduced the observed SST variability ranging from sub-seasonal to inter-annual.

observation of SST from satellites close to the coast and in a region notorious for its high cloud coverage. While (a) and (b) and intrinsic limitations of the model and the satellite product respectively, (c) is further explored in section **??** where we evaluate

the Moana Ocean Hindcast results against coastal temperature stations. Indeed, regions of larger RMSE agree in general with areas of large variability as shown in the SSH variance map in Figure 3.

The errors for the SSH and SST are summarized in Table 3.The Moana Ocean Hindcast shows a very good agreement with the observational products, especially for a non-assimilating simulation. This simulation presents similar errors for the SSH and out-performs data-assimilating global simulations for SST around New Zealand. The errors in the global simulations,

including GLORYS, are described by Souza et al. (2020). As presented above, the SSH errors must be taken with care since the Moana Ocean Hindcast simulation includes tides and inverse-barometer effect that are not included in the GLORYS reanalysis and are removed from the satellite data prior to the generation of the gridded product. Therefore, the differences are at least in part due to the improved physics. This is evaluated in detail when we compare the model results against tide gauge elevations in section 3.4.1.

**Table 3.** Summary of the deviations for the SSH (m) and SST ($^{o}$C) of the Moana Ocean Hindcast simulation compared to the AVISO and the OISST observational gridded products, respectively. Statistics for the root-mean-squared-error (RMSE), mean-absolute-error (MAE) and maximum-absolute-error (MaxAE) are presented.

| Variable | RMSE | MAE | MaxAE |
|----------|------|------|-------|
| SSH | 0.11 | -0.04 | 0.25 |
| SST | 0.23 | 0.18 | 1.53 |





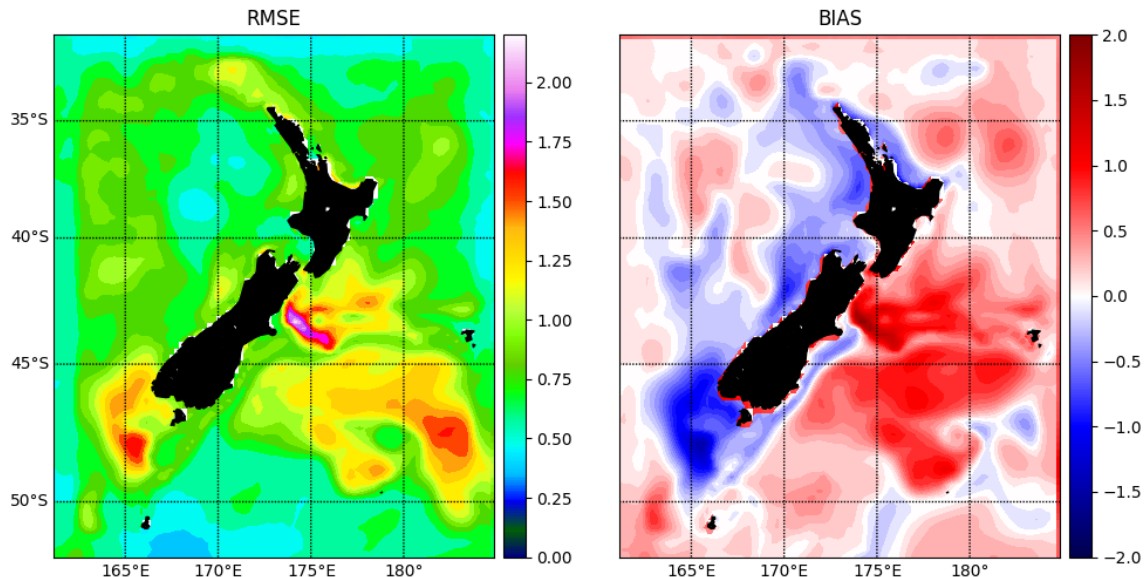

**Figure 6.** Root-mean-squared-error (RMSE) and BIAS of the Moana Ocean Hindcast SST ($^o$C) in comparison to the OISST observational product. It shows that differences between the simulation results and the OISST are concentrated in the locations of strong fronts and the coast. These relate to the fact this is a free-running simulation, differences in resolution, and the inability of the observational product in representing temperatures close to the shoreline.

## 3.2 Water column

We compare the daily mean fields from the Moana Ocean Hindcast model results against all the vertical profiles in the CORA 5.2 dataset. A total of 118040 temperature and 54787 salinity profiles were used in the model evaluation. These are unevenly distributed in time and across the model domain. Therefore, only aggregated information and scatter maps are presented.

The RMSE for both temperature and salinity (Figure 7) show an intensification near the surface - in particular in the top 20 m. Such intensification is related to the surface fluxes provided by the atmospheric simulation (CFSR) used to force the Moana Ocean Hindcast. The error decreases steadily with depth, with values in general under 1 $o$C for temperature and 0.15 g/kg for salinity bellow the mixed layer. These compare well to with the GLORYS errors presented by Souza et al. (2020).

Looking at the difference maps in Figure 8 one can see a general pattern of warmer and saltier waters to the east of New Zealand, and the opposite to the west. As shown in the RMSE profile, the differences are larger closer the surface. Although the difficulty in asserting the reasons behind such differences, there seems to be in part related to the surface forcing from CFSR and in part due to the boundary conditions from GLORYS. Souza et al. (2020) shows a similar pattern of differences for GLORYS in the thermocline and deep waters.



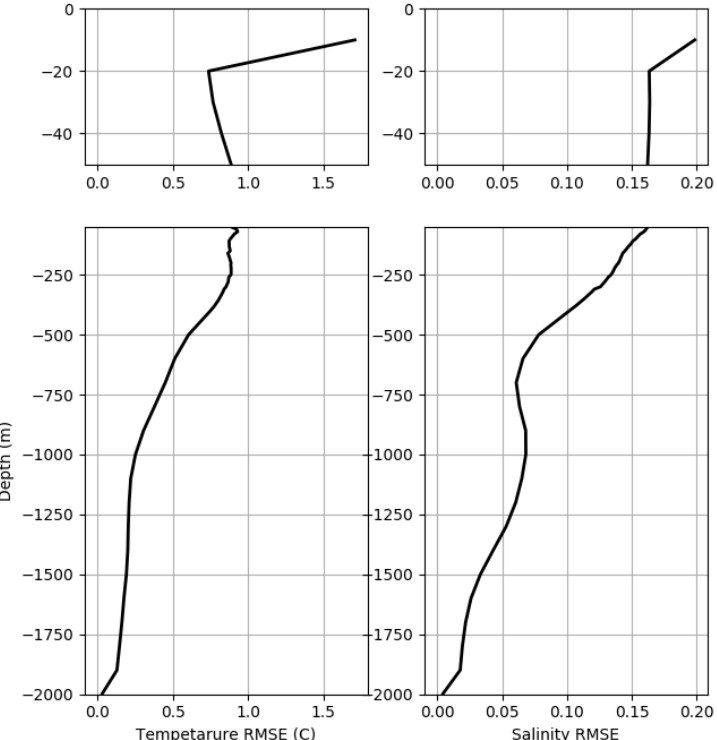

**Figure 7.** RMSE profiles for temperature ($^{o}$C) and salinity (g/Kg) of the Moana Ocean Hindcast simulations in relation to the CORA5.2 observations. A zoom in of the first 50m where the larger differences are observed is provided in the upper row.

Modelled and observed surface mixed layer depths (MLD) are compared seasonally and spatially over the hindcast period (Figure 9). Surface MLDs are estimated for individual CORA5.2 temperature profiles and Moana Ocean Hindcast temperature fields interpolated (nearest-neighbour in time, linear horizontally and vertically) onto the CORA5.2 profiles, with MLDs detected using a temperature difference criterion of 0.2 $^{o}$C (de Boyer Montégut et al., 2004) and a Matlab implementation of the Holte and Talley (2009) MLD algorithm available from http://mixedlayer.ucsd.edu/.

Results from the MLD analysis of the CORA5.2 observations (Figure 9, first column) are consistent with the literature and the expected seasonal dynamics of MLD thickness (Holte et al., 2017). During austral summer, MLDs across the region 31-45°S are shallow at < 25 m and the variability, indicated by 5th and 95th percentiles, is low (Figure 9, last column) with both metrics increasing toward higher latitude. Seasonal thickening of the MLD and increased variability in MLDs is evident across the entire domain with maximum MLDs reached during austral winter. The deepest MLDs and highest MLD variability is seen





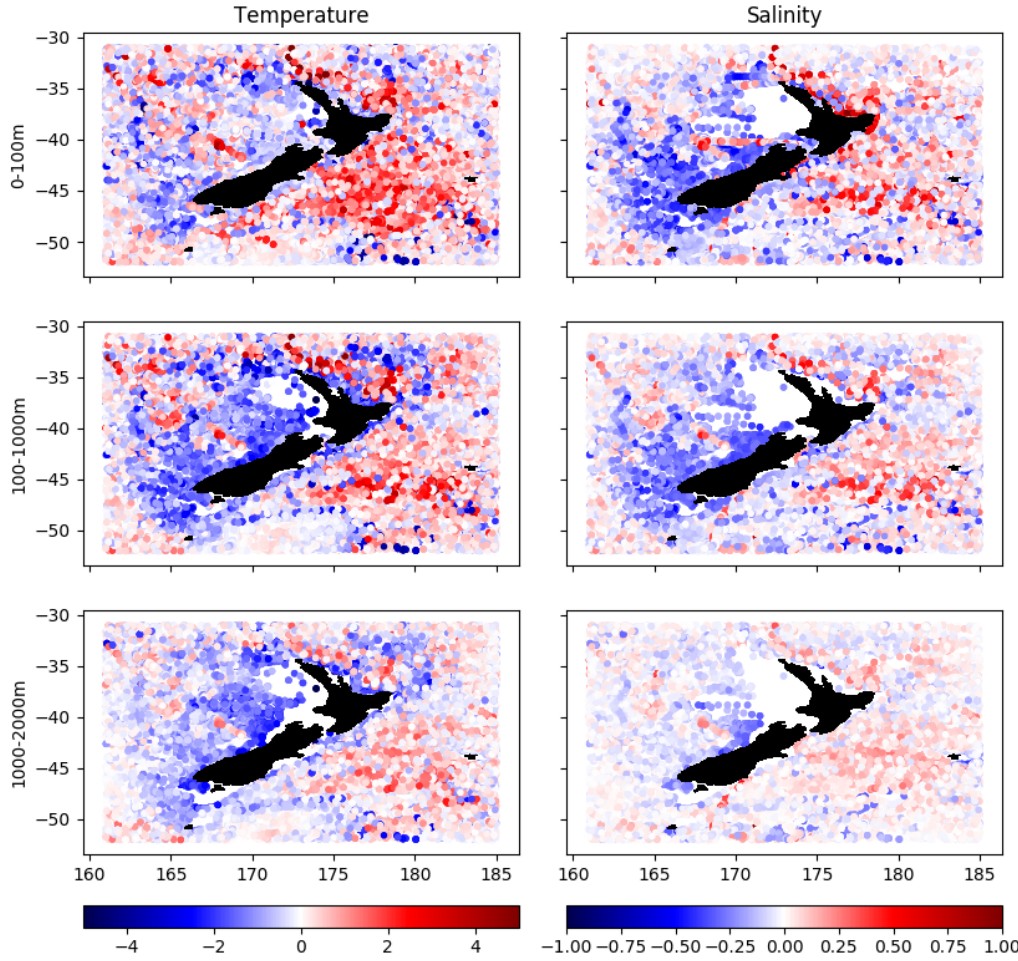

**Figure 8.** Scatter map of the depth mean deviation (by layers) of the Moana Ocean Hindcast results from the CORA5.2 temperature ($^o$C - left) and salinity (g/Kg - right) observations. The differences are divided by slabs corresponding roughly to the mixed layer(0-100m - upper row), thermocline (100-1000m - middle row), and deep waters (1000-2000m - lower row). A geographic distribution pattern is evident in the model results differences, that follow the same overall distribution presented by the GLORYS reanalysis.





south of 45°S, particularly along the borders of the Campbell Plateau and northern limit of the Antarctic Circumpolar Current, where MLDs exceed 250 m.

The model (Figure 9, second column) reproduces well the seasonal and spatial pattern in MLDs across New Zealand seen from CORA5.2, although there are some notable differences. These are evident by comparing differences between the model and corresponding observed MLDs over the entire domain (Figure 9, third column). The model generally under-estimates the MLD, with a domain-wide mean difference of 7-12 m, depending on season. The most notable differences are present in a halo around New Zealand during austral winter and spring, as well as in the vicinity of the Campbell Plateau. A comparison between

the 5th and 95th percentiles around the zonal mean MLDs from the model and observations (Figure 9, last column) provides a further assessment of the models performance in capturing temporal variability of the MLD. Generally, the model MLD variability falls within the envelope of the observed MLD variability for all latitudes and season; however, the 95th percentile of model MLDs is consistently lower than found in observations, suggesting that the model is under-estimating the depth of the deepest mixed layers in all seasons. We note that the accuracy of the daily MLDs is important for a range of applications,

for example when diagnosing drivers of marine heatwaves **?**.

### 3.3 Boundary Current Transport

The mean surface currents from the Moana Ocean Hindcast (Figure 10a) represent NZ's major boundary currents as described in Chiswell et al. (2015); Stevens et al. (2019). Current variability is greatest over the eddy-dominated regions where the EAUC separates from the coast (off North Cape, East Cape and Wairapapa), while the more coherent Southland Current shows little

directional variability (Figure 10b). To quantify NZ's major boundary current transport and variability we choose eight shore normal sections where the flow is maximum (Figure 10a, sections 1-8) and four sections where major boundary currents turn offshore (Figure 10a, sections A-D).

The volume transport through each section is computed daily and is given by,

$$Trans = \frac{1}{10^6} \int\limits_{-D}^{0} \int\limits_{x_0}^{x_i} (\mathbf{v}) dx dz, \tag{1}$$


where $x_0$ to $x_i$ is the cross-section distance and $-D$ is the depth of the section, $\mathbf{v}$ is the daily-averaged across-section velocity and the transport has units of Sv (1 Sv = $10^6$ m$^3$ s$^{-1}$). We choose the cross sectional area over which to compute current transport based on the mean alongshore velocity sections. The section length, starting from an inshore point at the shoreline, and depth are defined by the $-$ or $+$ 0.05 m $s^{-1}$ contour (the sign depending on the mean flow direction) in the

velocity mean, where the offshore reference point (OP) is set. In cases where the current core is not well defined (*i.e.,* Fiordland Current, Westland Current and west coast of New Zealand), a distance of 200 km offshore is chosen as the OP. The means and standard deviations of the daily volume transport over the long-term simulation, and the distances and depths over which





**Figure 9.** Seasonal mixed layer depths (MLD) computed from temperature profiles from CORA-5.2 (first column), the Moana Ocean Hindcast (second column) and the difference between these (third column) within the region 161-185°E, 31-52°S over the period 1994-2017 using a temperature threshold method (Holte and Talley, 2009). The last column indicates the zonal mean MLD from the Moana Ocean Hindcast (black solid) and CORA-5.2 (red solid), together with the 5th and 95th percentiles (shaded). Also shown are the number of temperature profiles in each latitude band (magenta solid).



transport is computed, are presented in Table 4. The Cook Strait section in our model is ≈ 15 km wide (represented by only 3
grid cells), compared to, in reality, a 22 km wide strait at its narrowest region.

**Table 4.** Alongshore transport (Sv) through cross-shore sections (Figure 10, sections 1-8) and the offshore sections (Figure 10, sections A-D) computed daily for the 25-year hindcast. The section length (which corresponds to the distance offshore for section 1-7) and depth over which the transport is computed is defined by the $-$ or $+$ 0.05 m s$^{-1}$ contour in the velocity mean (the sign depending on the mean flow direction), expect in cases where there is no defined core in which case a distance of 200km offshore is chosen. Section length and depth are included in the table. FD = Full Depth.

| | Mean (Sv) | Stdev (Sv) | Length (km) | Depth (m) |
|---|---|---|---|---|
| 1. East Auckland Current | 10.2 | 5.71 | 264 | 750 |
| 2. East Cape Current (Nth) | 28.0 | 8.40 | 151 | FD |
| 3. East Cape Current (Sth) | -37.6 | 10.2 | 279 | FD |
| 4. west coast of North Island | 3.57 | 3.65 | 200 | FD |
| 5. Southland Current | 9.32 | 2.66 | 122 | FD |
| 6. Fiordland Current | -4.32 | 12.4 | 200 | FD |
| 7. Westland Current | 0.0240 | 1.64 | 200 | FD |
| 8. Cook Strait | 0.19 | 0.50 | 15 | FD |
| 1. North Cape Separation | 14.7 | 12.3 | 148 | FD |
| 2. East Cape Separation | 24.7 | 14.9 | 131 | 2050 |
| 3. Wairarapa Separation | 42.0 | 11.1 | 150 | FD |
| 4. Southland Current Separation | 10.8 | 4.79 | 149 | FD |

We evaluate the model's ability to reproduce the large scale circulation through long-term averaged volume transport comparison with estimates presented in the literature. We limit this model assessment to three sections, relative to the EAUC, ECC and SC, given the extensive field work carried to date along the eastern margin of NZ continental shelf. Volume transport calculations are sensitive to the dataset used (e.g *in situ*, altimetry or model) and the section area over which transport is computed, which is directly dependent on data availability and/or assumptions made in the definition of the boundary current spatial extent (*e.g.,*horizontal and vertical). Nevertheless, comparing *in situ* versus modelled transport estimates allows for a reasonable quantitative assessment of the model's representation of the boundary currents.

Overall, modelled mean volume transport estimates from the main boundary currents are in agreement, within the range of the standard deviation, with values presented in the literature, indicating that the model reproduces the flow structure and magnitude of NZ's major boundary currents with a good degree of accuracy. Following is a more detailed description of how each boundary current compares with previous *in situ* and/or remote based volume transport estimates.

**East Auckland Current**

The mean modelled transport estimated for the EAUC, north-east of North Cape (Figure 10, sections 1), is 10.2 ± 5.71 Sv.



**Figure 10.** Mean surface current speed and velocity vectors (a) velocity variance ellipses (b). Data is from the daily-average output from the 25-year MOANA hindcast. Sections for the transport calculations are shown.





Our mean transport is found to be within the range of those reported in Roemmich and Sutton (1998) (9.0 Sv) and Stanton and

Sutton (2003) (9.5 ± 5.5 Sv), derived from XBT climatology and altimetry, respectively. Similar values were also encountered

by Fernandez et al. (2018) in the region using significantly longer data set. Their results show values of 12.4 ± 4.5 Sv and 12.6

± 2.6 Sv derived from 21 years of altimetry and 28 years of XBT measurements, respectively, and 8.4 ± 6.2 Sv from CTD

casts along same altimeter track. Those values are within range of 8 - 15 Sv derived from Argo float trajectories in the same

region (Bowen et al., 2014).

### East Cape Current (Sth)

The mean modelled transport estimated in the ECC South region (Figure 10, sections 3) is 37.6 ± 10.2 Sv. This estimate is

considerably higher than those presented in the literature, however key difference in the calculation methods and locations

exist. The ECC transects of Fernandez et al. (2018) are to the north and to the south of our chosen transect and estimate

altimeter-derived mean and standard deviation of volume transports of 10.5 Sv (2.7 Sv) and 5.6 Sv (2.2 Sv), respectively. The

transect that is further to the north (directly off East Cape) is located where current velocities are considerably lower, while

the transect to the south is downstream of the peak current velocity, transverses the equatorward counter current and does not

extend offshore into the core of the ECC (Figure 10). In contrast, our section was chosen where the ECC (south of East Cape)

shows the strongest velocities, and transport is considerably strengthened due to recirculation of the Wairarapa eddy. This

strengthening due to recirculation is also seen in the EAC (**?**). This study is indeed the first study that we are aware of that has

estimated transport in the ECC at this latitude, where velocities are strongest. Furthermore our transport estimate encompasses

the entire cross-section through the current (based on the 0.05 m s$^{-1}$ mean velocity contour) extending 279 km offshore and

to the full water depth (below 3000m). Chiswell (2005) estimate the transport in the ECC feeding the Wairarapa eddy to be 15

Sv relative to the 2000 dbar, yet they note that this is likely to be an underestimate as the current core extends deeper than the

2000 dbar (Chiswell, 2003).

### Southland Current

The mean modelled transport estimated in the SC region (Figure 10, sections 5) is 9.32 ± 2.66 Sv . Similar values (10.4

Sv) have been reported by Chiswell (1996), inferred from geostrophic velocities estimated from a 1-year long CTD survey

conducted along a transect off Oamaru (virtually the same location as of the section adopted here). These values are also in

agreement with those found by Sutton (2003) (8.3 ± 2.7 Sv) obtained from full-depth transport estimates derived from CTD

survey carried between years 1993 and 2000 over a region offshore Otago Peninsula encompassing the north of Campbell

Plateau and south of Chatham Rise. More recently, Fernandez et al. (2018) derived the SC volume transport from 1993-2012

altimeter data across two sections, south and north of our reference section, reporting 7.2 ± 0.8 Sv and 10.6 ± 1.0 Sv, respec-

tively.

### Cook Strait

We also assess the mean cross-sectional transports at the Cook Strait (Figure 10, sections 8) given the significance and role





that volume exchange across the strait plays in the upper water column ocean circulation in the central New Zealand region.
The mean modelled transport across the Strait is $0.19 \pm 0.50$ Sv. The high standard deviation relative to the mean illustrates
the variable nature of the residual transport, and it can be expected that the mean transport is sensitive to the time period over
which the mean is taken. Stevens (2014) estimated a mean transport of 0.25 Sv based on residual (low-passed filtered at 48
hours) currents from 20-month continuous ADCP measurements. Hadfield and Stevens (2021) estimate a 3-year mean volume

flux of 0.42 Sv $\pm$ 0.08 Sv based on modelled-measured adjustments. Our modelled value may be lower as the Cook Strait
width is 15km at its narrowest point in the model, compared to the real width of 22 km.

## 3.4 Coastal sea level and water temperature

### 3.4.1 Coastal sea level

Observed and modelled sea level variability are compared over a 3 year period (Jan 2015 - Dec 2017). Data from fifteen oceanic
grid locations adjacent to the coincident LINZ stations are extracted from the Moana Ocean Hindcast. Sea level observations
from the LINZ tide gauge observations are hourly averaged to match hourly model output sea surface height ($\zeta$) from the
ROMS hindcast. The software T_TIDE (Pawlowicz et al., 2002) is used to conduct harmonic analysis, extracting the amplitude
and phase from the 8 largest tidal constituents in both observations and the model output.

Results from harmonic analysis for the four largest tidal constituents (three semidiurnal ($M_2$, $S_2$, $N_2$) and one diurnal ($K_1$)
constituent, Figure 11), reproduce the well-documented spatial structure of tidal amplitude around New Zealand (e.g., Walters
et al., 2001; Lane et al., 2009; Stevens et al., 2019). Most tidal constituents are amplified towards the northern portions of the
North Island, with smaller sea sea level variability over the South Island. Both observations (red dots, Figure 11, left panels)
and the model (black dots, left panels) demonstrate this variability. Despite LINZ tidal stations located in both harbour and
open-coast locations, the spatial variability in tidal constituents is well-reproduced in the Moana Ocean Hindcast. In addition
to amplitude, the phase progression around the north and south islands is also well-reproduced in both the semidiurnal (Figure
11 B, D, F) and diurnal tidal band (Figure11, H).




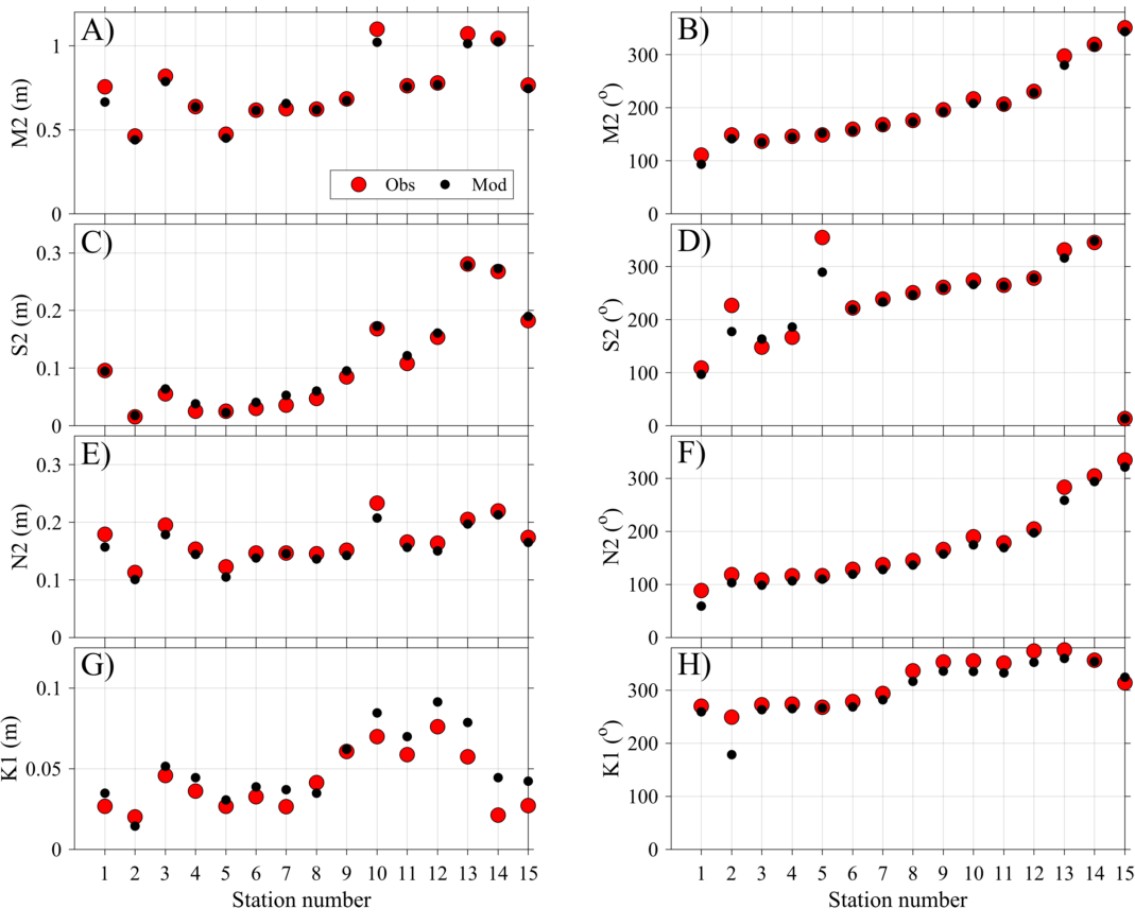

**Figure 11.** Harmonic tidal amplitude (left panels, A - G) and phase (right panels, B - H) estimated from observed sea level (red) and modeled (black) time series for 15 stations around New Zealand (Table 1). Station numbers (1 - 15) are indicated to follow counter-clockwise Kelvin wave propagation from the southeast station (Port Chalmers, 1) to the southwest (Purseygur, 15). Amplitude and phase are shown for the $M_2$ (A, B), $S_2$ (C, D), $N_2$ (E, F), and $K_1$ (G, H) tidal constituents.

A summary of all 8 analysed tidal constituents across all 15 stations is presented as a Root-Mean-Square-Error (RMSE) between model and observations (e.g., Wang et al., 2009), separately for amplitude and phase (Table 5). In general, the RMSE amplitude is an order of magnitude smaller than the Root-Mean-Square (RMS) of the tide gauge observed amplitude for all semidiurnal constituents. The largest RMSE amplitude (4 cm), the $M_2$ tide, is also the largest tidal constituent measured at the tide gauges. The RMSE for $O_1$ and $K_1$ diurnal constituents were small (1 cm), and represent 1/3 and 1/5 the amplitude of the observations respectively. The RMSE of phase error was also smaller for the large amplitude semidiurnal constituents compared to diurnal constituents. The largest semidiurnal (diurnal) constituent error, $K_2$ ($P_1$) had an RMSE phase of 1.2 (4.4) hours, perhaps indicating an accumulated error due to harbour propagation unresolved in the Moana Ocean Hindcast.



**Table 5.** Collected tidal amplitude and phase statistics at the 15 tide gauge stations. Root-Mean-Square (RMS) observed amplitude for the 8 largest tidal constituents. Root-Mean-Square-error (RMSE) in amplitude and phase between observations and hindcast model.

| Const | RMS Obs amp (m) | RMSE amp (m) | RMSE phase ($^o$) |
|-------|-----------------|--------------|-------------------|
| $M_2$ | 0.77 | 0.04 | 7.6 |
| $S_2$ | 0.13 | 0.01 | 22.9 |
| $N_2$ | 0.17 | 0.01 | 14.0 |
| $K_2$ | 0.04 | 0.004 | 34.9 |
| $O_1$ | 0.03 | 0.01 | 16.5 |
| $K_1$ | 0.05 | 0.01 | 22.8 |
| $P_1$ | 0.01 | 0.003 | 65.6 |
| $Q_1$ | 0.01 | 0.003 | 56.5 |

Tides account for $> 98\%$ of the variance in sea level variability at all observed and modelled stations. However, non-tidal sea level (SLA) fluctuations can be an important indicator of such oceanic processes as storm surge, wind-driven up/down-welling, and geophysical Kelvin and Rossby waves (e.g., Walters et al., 2001; Lane et al., 2009). Although a thorough decomposition of coastal SLA into various forcing mechanisms is beyond the scope of the current study, a comparative analysis between observations and modeled SLA is performed here to indicate the extent to which natural SLA variability around New Zealand

is produced in the ROMS Ocean Hindcast. Model and observed SLA time series from 3 locations roughly covering the New Zealand latitude span, Manukau ($SLA_{TG13}$, Figure 12 A), Wellington ($SLA_{TG5}$, Figure 12 B) and Port Chalmers ($SLA_{TG1}$, Figure12 C) are displayed for 2017. The coastal SLA is calculated from a 40-hour low-pass filter of detided, hourly time series. Model-data time series are compared with Wilmott Skill (Willmott, 1981),

$$\text{WS} = 1 - \frac{\text{MSE}}{\langle (|m - \langle o \rangle| + |o - \langle o \rangle|)^2 \rangle} \tag{2}$$

where $m$ ($o$) is the modeled (observed) subtidal sea level, angle brackets denote a time-mean and the mean square model-data difference (error) is denoted $\text{MSE} = \langle (m - o)^2 \rangle$. The high values of Wilmott Skill ($\text{WS} > 0.9$) at the three locations demonstrate that sub-tidal sea level signals across a variety of events and time scales are well-reproduced across New Zealand (Figure 12). Although 11 of the 15 stations have $\text{WS} > 0.9$, a few comparisons were less favorable. The northern-most location for example, North Cape ($SLA_{TG12}$, not shown) had $\text{WS} = 0.72$, perhaps indicating that some observed features of coastal sea

level variability require higher-resolution modelling than currently simulated by the Moana Ocean Hindcast.



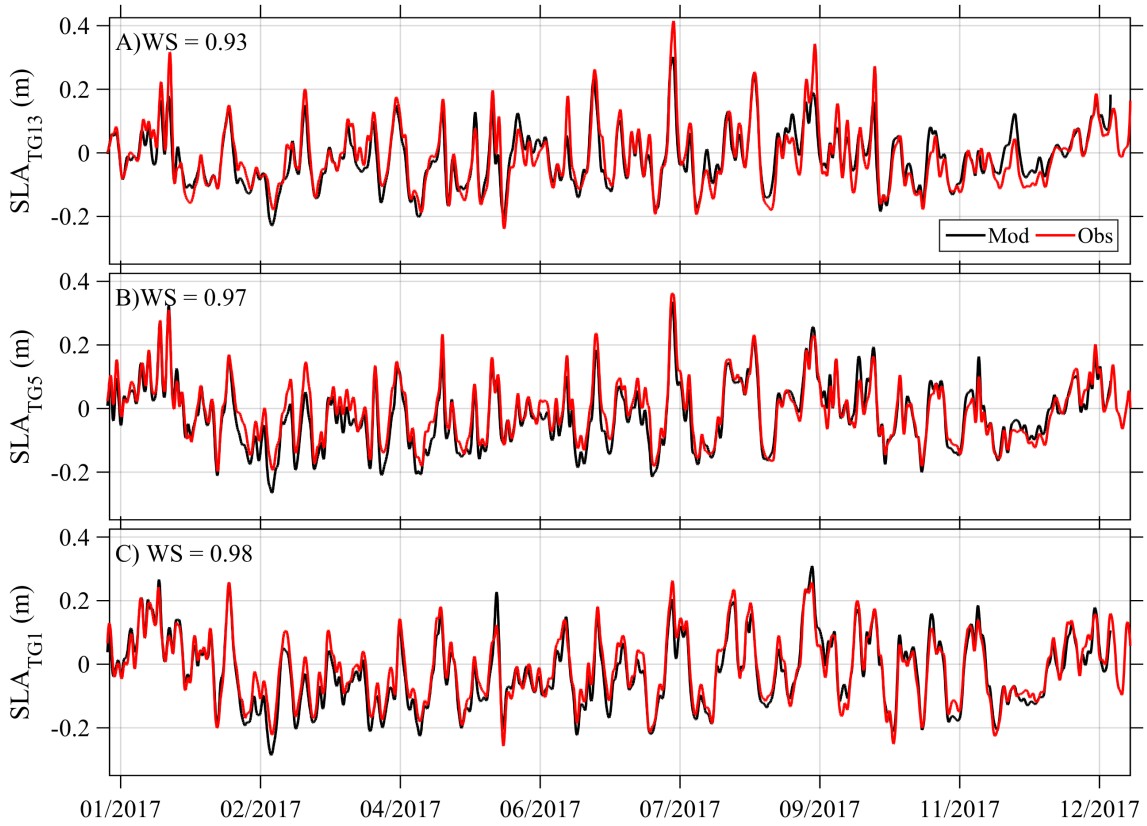

**Figure 12.** Observed (red) and modelled (black) subtidal coastal sea level anomaly for calendar year 2017 at 3 stations, A. Manukau ($SLA_{TG13}$), B. Wellington ($SLA_{TG5}$) and C. Port Chalmers ($SLA_{TG1}$)

### 3.4.2  Coastal daily water temperature

Observed and modelled daily water temperature are compared over the hindcast period at 10 available temperature stations spanning the latitudinal range of New Zealand (Figure 13, sites shown in Figure 1). At all stations, the seasonal cycle of temperature is large relative to other sources of variability. Differences in observed temperature between stations likely reflect
a combination of the station latitude, exposure to the various boundary currents around NZ and that some coastal sampling stations are located in shallow bays or harbours. The length of the available observed time series for model-data comparison varies considerably with location (Table 2), therefore in this analysis primary statistics are presented for observations that overlap in time with the Moana Ocean Hindcast. This period varies in length from the entire hindcast period (e.g., Evans Bay and Portobello, Figure 13 G, I) to as little as a 4-year period at Bluff (Figure 13 J).





**Figure 13.** Observed (red) and modelled (black) daily coastal sea surface temperature from 10 stations around New Zealand (Table 2) roughly ordered from northern-most station (A. Ahipara) to southern-most (J. Bluff) as shown in Figure 1.




The primary time series statistics compared between observed and modelled temperature are the time-mean, amplitude of the
seasonal cycle, and the standard deviation of daily temperature anomaly $\sigma_{\mathrm{obs}}$ and $\sigma_{\mathrm{mod}}$ (Figure 14). The anomaly is calculated
as the difference between the raw daily individual time series minus a harmonic regression fit consisting of the time-mean,
seasonal cycle, and the first two higher harmonics of the seasonal cycle. The time-mean temperature at each station is very well
reproduced by the model hindcast and is dominated by the north-south latitudinal gradient of the coastal temperature (Figure

14 A). The largest mean difference (bias) is a cold model bias found at the Tauranga wave buoy ($-0.57^o$C, latitude $-37.7$). Note
that this measurement is taken from the base of a wave buoy (Table 2), a different method from the other stations. Due to a
lack of wave buoy sampling during winter conditions, the Tauranga measurement is also slightly skewed towards summertime
measurements.

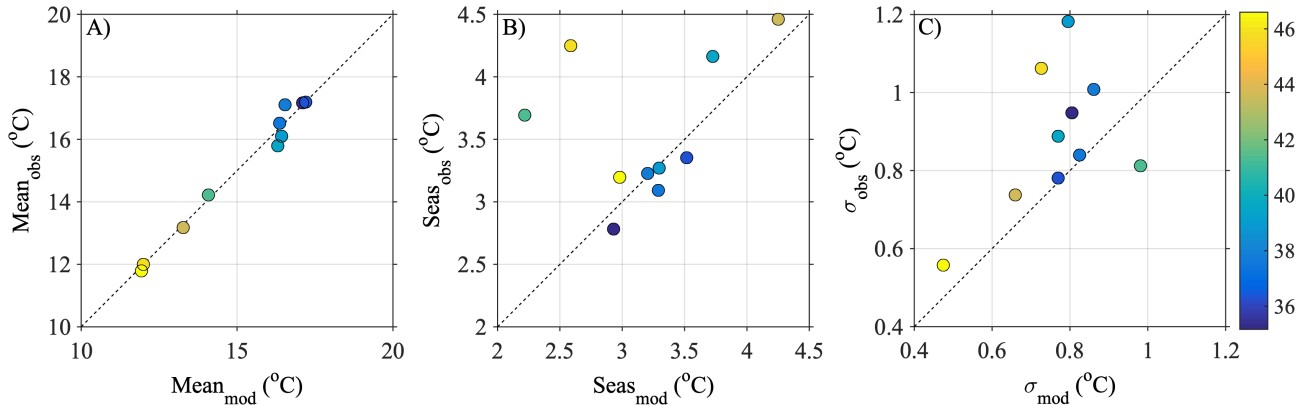

**Figure 14.** Primary statistics of coastal water temperature in model (x-axis) and observations (y-axis). A. Time-mean temperature, B. Amplitude of seasonal harmonic, C. Standard deviation of daily temperature anomaly. Colours denote latitude of coastal measurement station location.

The amplitude of the seasonal cycle varies across station locations between 2 - $4.5^o$C, consistent with previous results

(Chiswell and Grant, 2019). The amplitude of the modelled seasonal cycle falls within $0.25^o$C of a 1:1 line for seven out of
the ten stations, with little discernible preference for latitude (Figure 14 B). The three stations with the largest discrepancy in
seasonal cycle are all under-reproduced in the model. With the difference from 1:1 listed in descending order, these locations
are Portobello ($1.66^o$C), Evans Bay ($1.47^o$C) and Napier ($0.43^o$C). These locations are all located within semi-enclosed bays
and harbours where a larger seasonal cycle can be observed but is potentially due to land-air-sea processes unresolved in

a regional-scale oceanic model of this resolution. Satellite-derived annual cycles show coastal regions around New Zealand
differ from $\approx 3^o$C in northern New Zealand to about $\approx 1^o$C in southern New Zealand (e.g., Wijffels et al., 2018). The coastal
annual cycles presented here, are typically higher than the satellite derived cycles, consistent with previous analysis of coastal
stations (Chiswell and Grant, 2019).

Non-seasonal, daily temperature anomaly variability ranges between 0.4 - 1.2 $^o$C in both model and observations (Figure 14

C). At all locations, $\sigma_{\mathrm{obs}}$ is larger than $\sigma_{\mathrm{mod}}$, except in Evans Bay, Wellington which is not fully-resolved by the 5 km grid
spacing. Overall, these coastal temperature anomalies show a decrease with latitude in both observations and model, future





close inspection of these time series are needed to understand this feature. In addition to the primary temperature statistics, the daily temperature anomaly time series are further compared with cross-correlation coefficients and Willmott Skill (Eq.2) between the model and observations. In general, both metrics are high and significant at the 95% confidence level (Table 6) indicating that the processes regulating temperature anomalies at these stations are represented in the Moana Ocean Hindcast. Locations with somewhat lower correlations are similar to those with large differences in $\sigma_{\mathrm{obs}}$ compared to $\sigma_{\mathrm{mod}}$ (Figure 14 C).

**Table 6.** Coastal daily surface temperature model-data comparison statistics. Wilmott Skill is used as the model hindcast predictability metric, Pearson correlation coefficient used as degree of correspondence.

| SST station | Wilmott Skill | Correlation |
|---|---|---|
| A. Ahipara | 0.85 | 0.73 |
| B. Leigh | 0.87 | 0.76 |
| C. Moturiki | 0.87 | 0.75 |
| D. Tauranga | 0.86 | 0.79 |
| E. New Plymouth | 0.83 | 0.75 |
| F. Napier | 0.77 | 0.64 |
| G. Evans Bay | 0.76 | 0.59 |
| H. Lytleton | 0.86 | 0.75 |
| I. Portobello | 0.70 | 0.53 |
| J. Bluff | 0.88 | 0.79 |

## 4 Conclusions

Our rigorous model evaluation shows that the **Moana Ocean Hindcast** provides a consistent, continuous and realistic representation of the ocean state around New Zealand. It includes important physical processes usually absent from global simulations, such as tides and the inverse barometer effect, the contribution from all the main rivers and a more detailed and realistic bathymetry. The results are available at higher spatial and temporal resolutions than most open access datasets, providing an optimal basis for a series of analysis of the ocean dynamics in this region.

The model results outperform the global models in the coastal region. The multi-decadal time frame of the simulation makes it useful for rigorous statistical analysis, including extreme value analysis necessary for coastal infrastructure projects. The simulation represents well the ocean variability at a range of time scales from a few hours to inter-annual. This makes the present configuration a good starting point for regional climate downscaling studies since it does not present intrinsic biases related to internal processes.

This first multi-decadal, high resolution, open access model represents a significant step forward for ocean sciences in New Zealand.



*Code availability.* The "Regional Ocean Model System" (ROMS) has a large user base. Access to the source code, model documentation, and can discussion forum is available at 'https://www.myroms.org/'. The Moana Ocean Hindcast configuration files and ROMS model source code used in this simulation are available at https://github.com/joaometocean/moana_hindcast.git

*Data availability.* The Moana Ocean Hindcast model output is available at 'http://thredds.moanaproject.org:8080/thredds/dodsC/moana/ocean/',
and is directly citable (Souza, 2022). Open access to the GLORYS reanalysis is provided by Copernicus Marine Environment Monitoring Service (CMEMS) (https://resources.marine.copernicus.eu/product-detail/GLOBAL_MULTIYEAR_PHY_001_030/INFORMATION).

All observations used in the present study are publicly available.

CMEMS products are available upon registration. The link to the sea surface height satellite product is available at ftp://my.cmems-du.eu/
Core/SEALEVEL_GLO_PHY_L4_MY_008_047/cmems_obs-sl_glo_phy-ssh_my_allsat-l4-duacs-0.25deg_P1D and to the CORA5.2 in-
situ observations at https://resources.marine.copernicus.eu/product-detail/INSITU_GLO_TS_REP_OBSERVATIONS_013_001_b/INFORMATION.

NOAA High Resolution SST data provided by the NOAA/OAR/ESRL PSL, Boulder, Colorado, USA, from their Web site at https://www.ncei.noaa.gov/data/sea-surface-temperature-optimum-interpolation/v2.1/access/avhrr/.

The observations from the coastal sea level stations can be accessed at https://www.linz.govt.nz/sea/tides/sea-level-data/sea-level-data-downloads. University of Auckland Leigh Marine Laboratory seawater temperature is available through the NOAA National Oceanic Data Center
https://www.nodc.noaa.gov/archive/arc0075/0127323/.

The compiled version of the coastal station temperature observations and corresponding model data are available at https://zenodo.org/record/6399921#.YkTKhnVBxhE.

*Author contributions.* Joao M. A. C. de Souza lead the development of the Moana Ocean Hindcast, was responsible for the experiment execution, and is the main author of the present manuscript. Sutara H. Suanda co-ordinated the data retrieval, analysis and writing of coastal
sea level and water temperature. Phellipe P. Couto elaborated the discussion of the volume transport estimates obtained in this work in comparison with those presented in the literature. Robert Smith conducted the mixed layer depth analysis. Colette Kerry used the hindcast to characterise NZ's boundary current circulation. Moninya Roughan conceived the idea and obtained the funding. All authors contributed to the model analysis, writing and reviewing the manuscript.

*Acknowledgements.* This work is a contribution to the Moana Project (www.moanaproject.org), funded by the New Zealand Ministry of
Business Innovation and Employment, contract number METO1801. The Salto/Duacs altimeter products and the CORA dataset were produced and distributed by the Copernicus Marine and Environment Monitoring Service (CMEMS) (http://www.marine.copernicus.eu). Bathymetric data was provided by GEBCO: GEBCO Compilation Group (2021) GEBCO 2021 Grid (doi:10.5285/c6612cbe-50b3-0cff-e053-6c86abc09f8f). We acknowledge Phillip Sutton and Steve Chiswell (NIWA) for gracious access to the daily coastal water temperature record and Land Information New Zealand (LINZ) for providing open access sea level data to the public. Modifications to the model con-
figuration were made after discussions with several Moana Project partners. In particular, we would like to acknowledge the important





contributions from Mark Hadfield, John Wilkin, Brian Powell, and Gary Brassington. Such discussions were a driving force for continuous improvement of the results.



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
