# Peer review of "Moana Ocean Hindcast - a 25+ years simulation for New Zealand Waters using the ROMS v3.9 model."

_EGUsphere, 2022_

## Referee Comment (RC1)

**General comment:**
The manuscript entitled "Moana Ocean Hindcast – a 25+ years simulation for New Zealand Waters using the ROMS v3.9 model" by Souza et al. presents an extensive model-data comparison of a high-resolution ocean model for the New Zealand region. The comparisons highlight the improvements of the high-resolution simulation over the global data-assimilative model used for boundary conditions. The manuscript is well-structured, moving from the large-scale to the coastal ocean, however it is my opinion that minor changes are needed before this manuscript can be considered for publication.

I appreciated the manuscript for being well-structured and for trying to present a substantial amount of information in a succinct way. However, in doing so some of the results aren't explored in any detail and some of the model-data comparisons are glossed over (e.g. SST). The Conclusion sections is a bit disappointing. This section could be improved by a) highlighting the strengths and short-comings of the model and b) by including some of the ways in which the model can be improved on.

**Specific comments:**

Page 1, Line 20: The authors state that the New Zealand fishing industry is expanding to include coastal aquaculture. New Zealand already has extensive coastal aquaculture activities based around three marine species: salmon, mussels and oysters. However, the Government's Aquaculture Strategy aims to a) expand and improve the existing aquaculture activities, b) advance land-based aquaculture and c) extend aquaculture into the open ocean by 2035. The aquaculture industry in New Zealand are already exploring moving further offshore (so-called open ocean aquaculture) due to coastal warming and the impacts of marine heat waves.

Page 2, Line 46: The statement "revolutionize the understanding and prediction of ocean processes in New Zealand" seems to be a bit of a bold claim. Regional models like the one described here are well-used to understand ocean processes in a range of environments. I agree with the author's underlying view that this simulation will significantly contribute to improve our understanding and predictability of ocean processes but to claim that it will be revolutionary is an exaggeration.

Page 2, Lines 48-49: What do the authors mean by "7 days forecasts daily"? Were daily forecasts created for a period of 7 days?

Page 3, Line 74: Please clarify what you mean by mass structure

Page 6, Line 104: Please be more specific with respect to the local sources that were used for the model bathymetry.

Page 6, Line 107: The authors mention that PGE associated bottom velocities were used to determine which grid cells required bathymetric smoothing. Since this is not a commonly used method in ROMS, please provide more details about this method.

Page 6, Line 118: You use Souza et al. (2020) to justify using CFSR for the atmospheric forcing. However, Souza et al. (2020) only compared the ocean component of CFSR to other ocean models and did not look at the atmospheric component. Is there any evidence that the atmospheric component of CFSR out-competes other atmospheric reanalysis products such as JRA-55 in the New Zealand region?

Page 10, Lines 208-209: Please be more specific with regards to the location of the high and low SSH centres and the large-scale currents hindicated by these SSH centres?

Page 12, Line 230: The authors note that it is interesting that the hindcast reproduced the 2018 temperature peak. This suggest that they didn't expect the model to reproduce this event. Please elaborate why this is particularly noteworthy.

Page 12, Line 234-235: The authors mention that the bias pattern is similar to that found for GLORYS in Figure 7 of Souza et al. (2020) but then don't give an explicit description of the biases in the Moana Ocean Hindcast shown in Figure 6.

Page 13, Lines 238-240: While (a) and (b) **are the result** of intrinsic limitations of the model. Also, the section number for the section addressing (c) is missing. Point (c) is listed as "issues related to the observations of SST from satellites close to the coast and in a region notorious for its high cloud coverage", yet Section 3.4.2. which I suspect is suppose to look at this limitation in more detail only looks at daily sea surface temperature at a number of coastal stations around New Zealand, and doesn't examine the limitations of satellite SST close to the coast.

Page 13, Table 3: RMSE and MAE are standard statistics used in model validation but are often calculated in different ways. Furthermore, maximum-absolute-error is not a commonly used statistic. I recommend that the equations for all the statistics presented in the manuscript are included.

Page 14, Line 251: Was the model output used for the comparison co-located in space and time with the temperature and salinity profiles or was the time-averaged model output extracted at the location of the profiles?

Page 14, Line 259-260: The phrasing makes this sentence hard to understand. I think the authors are trying to say something along the lines of "Despite the difficulty in asserting the reasons behind such differences, the differences seem to be in part related to the surface forcing from CFSR and in part due to the boundary conditions from GLORYS.

Page 17: The section on MLD really gives a nice description of what is observed in the observations and how the model differs from these observations. It would have been really good if this same approach was used throughout the manuscript.

Page 19, Lines 309-311: It is clear from the statement here that you are cognisant of the fact that volume transport calculation are sensitive to the horizontal and vertical extent over which it is calculated. Was any effort made to try and match the sections used here to the spatial extent over which transport has been calculated in the literature for a better comparison?

Figure 11: Can you provide any explanation for some of the large differences observed between the observations and the model (e.g. stations 10-15 in subplot G)?

Page 27, Line 425-426: "Satellite-derived annual cycles show coastal regions around New Zealand differ from…" This sentence is hard to understand due to the phrasing and requires some clarification.

Page 28, Line 431-432: Here you claim that additional analysis of the coastal SST time-series are needed to understand the latitudinal decrease in SST and this might certainly be the case. However, one contributing factor to the latitudinal decrease would be a change in the background surface water masses as one moves from north to south with warm sub-tropical water in the north and cooler sub-Antarctic water in the south.

Page 28, Line 444: "The model results outperform the global models in the coastal region". This is a very generalised statement that can't really be backed up by any of the results presented in the manuscript. The results presented here only compares the temporal mean SSH and variance of SSH of the Moana Ocean Hindcast to that of one global model, GLORYS.

**Technical corrections:**
Page 2, Line 24-25: Move the second set of references (e.g. Chaput et al. 2022; Silva et al., 2019) to the end of the sentence.

Page 2, Line 27: **an** important source

Page 2, Line 53: replace 'bibliography' with 'literature'

Page 3, Line 56: Change 'Marine Heat Wave' to marine heatwave as used throughout most of the manuscript.

Page 3, Line 57: Remove the quotation marks around "sea surface temperature"

Page 3, Line 59: Please provide the name of this western boundary current

Page 3, Line 68: Please define DA the first time that it is used. A reader familiar with ocean modelling and forecasting will know that it stands for Data Assimilation but someone new to the field might not necessarily know this.

Page 3, Lines 72-76: Please provide references to substantiate these statements.

Page 3, Line 74: **Three K**ings Islands

Page 3, Line 80: Change following publications to subsequent publications or future publications.

Page 3, Line 92: Unbold 5km resolution

Page 3, Line 94: Chatham **I**slands

Page 3, Line 94: **three** New Zealand main islands

Page 3, Line 95: **western** boundary

Page 3, Line 100: Change following development to subsequent development.

Page 6, Line 101: Unbold 50 vertical layers. Also, ideally a sentence should not start with numerical digits but if it does then the number should be written out.

Page 6, Line 122: **(**Lellouche et al. 2021**)**

Page 6, Line 123: 1 day$^{-1}$

Page 6, Line 128: Change warm-up period to spin-up period. Did the model reach a steady state after just 1 year's integration? I suspect that it did but it did.

Page 6, Line 132: global ocean **reanalyses**

Page 7, Line 144: **the** Mercator Ocean

Page 7, Line 150: can be determinant **factors** for the representation of the  sea surface height

Page 7, Line 160: The Moana Ocean Hindcast

Page 7, Line 186: Change forward to onward

Page 9, Line 195: collected by  New Zealand's

Page 9, Line 202: (NZ-OOS; **O**'Callaghan et al., 2019)

Page 10, Line 210: **es**pecially in the region

Page 10, Line 211: "constitutes in" change for something else.

Page 10, Line 215-217: Including a schematic representation of the currents around NZ would make it easier on the reader to follow along.

Page 12, Figure 4: In Figure 3, the first panel was the Moana Backbone followed by GLORYS and then AVISO.  It would make it easier to follow along if the order is kept the same between figures.

Page 12, Figure 4 caption: G**L**ORYS

Page 12, Lines 223-224: The phrasing makes this sentence hard to understand.

Page 12, Line 225: pattern is  similar

Page 12, Line 231: events with duration **on** the order

Page 13, Lines 243: Please rephrase this sentence so that it is clearer that SSH errors are similar to that from global simulations, while SST performs better. It will also help if the statistics for GLORYS is included in Table 3.

Page 14, Line 253: Please elaborate on 'the aggregated information' that is presented

Page 14, Line 256: with values  general**ly** under 1**º**C

Page 14, Line 257: compare well  with

Page 14, Line 259: As shown in the RMSE profile **(Figure 7)**

Figure 7 caption: A zoom  of the

Figure 8: At first glance it is a bit confusing to have the depth-ranges used to generate the subplots as the label on the y-axis when in fact the y-axis represents latitude.

Page 17, Line 273: Indicate the location of the plateau and other features mentioned in the manuscript on Figure 1.

Page 17, Line 281: model**'s**

Page 17, Line 285: It appears that a reference is missing here.

Page 17, Line 288: Chiswell et al. (2015) **and** Stevens et al. (2019)

Page 17, Lines 299-301: The phrasing of this sentence makes it very hard to understand. The description in Table 4's caption makes much more sense. This sentence should be rephrased so that it is easier to understand how the transport was calculated along the sections.

Table 4: The offshore sections need to be relabled A-D to match the sections in Figure 10 and west of section 4 should be capitalised. Also, the Cook Strait section is not indicated on Figure 10.

Page 19, Line 307: Change relative to corresponding

Page 19, Line 308: carried **out** to date

Page 19, Line 314: I would suggest including the transports reported in the literature for the different currents in Table 4 for easy comparison.

Page 19, Line 315: Change Following to Below

Page 19, Line 316: "remote based" I suggest changing it so that it is clearer that it is volume transport calculated from remotely sensed data.

Page 19, Line 318: Include a cross-reference to Table 4 for the transports reported here.

Page 21, Line 322: using **a** significantly

Page 21, Line 324: along **the** same
Page 21, Line 324-325: This sentence is hard to comprehend. I suggest rephrasing it to something like "These values are also consistent with a volume transport of 8-15 Sv derived from Argo float trajectories in the same region (Bowen et al., 2014)."

Page 21, Line 328: Add a cross-reference to Table 4 for the transports reported here.

Page 21, Line 330: The model volume transport calculated for the ECC south is significantly higher than that reported by Fernandez et al. (2018). How does the model volume transport compare to that of Fernandez et al. (2018) if calculated along similar sections used by Fernandez et al. (2018)?

Page 21, Line 356: It looks like a reference is missing here.

Page 21, Line 355: same location as **for** the section

Page 21, Line 348: carried **out** between  1993 and 2000

Page 21, Line 353: cross-sectional transport **through** the Cook Strait

Page 22, Line 356: Include a cross-reference to Table 4 for the transports reported here.

Page 22, Line 367: Everywhere else throughout the manuscript it is referred to as the Moana Ocean Hindcast. I suggest changing ROMS hindcast to Moana Ocean Hindcast to keep it consistent with your vocabulary.

Page 22. Line 369: Results from **the** harmonic analysis

Page 23, Line 377: "Root-Mean-Square-Error (RMSE)". Abbreviations should be written out at the first instance where they are used and not towards the end of the manuscript.

Page, 24, Line 390: Everywhere else throughout the manuscript it is referred to as the Moana Ocean Hindcast. I suggest changing ROMS hindcast to Moana Ocean Hindcast to keep it consistent with your vocabulary.

Page 25, Line 402: Was the modelled data extracted at the grid cell closest to the station locations?

Page 27, Line 421-422: The phrasing makes this sentence hard to understand.

Page 27, Line 424-425: I suggest rephrasing the second part of this sentence e.g. is potentially unresolved in a regional-scale oceanic model of this resolution due to land-air-sea processes.

Page 27, Line 431: This sentence would read better if it is split in to i.e. "...decrease with latitude in both observations and model. Future ..."

Table 6 caption: I recommend splitting the last sentence of the caption in two.

Page 28, Line 439: unbold Moana Ocean Hindcast

Page 28, Line 443: series of **analyses**

---

## Referee Comment (RC2)

General comment

Overall this is a good and detailed paper on a hindcast dataset while providing background information and validation of the dataset. The methods undertaken are robust and I commend the authors for their interesting study. As is mentioned in the manuscript, the work builds on earlier work and extensively compares model results with other data sources (like GLORYS). The work presented in novel and provides a major step in the direction of predictive models and I think it is worthy of publishing in NHESS. However, I have listed a couple of modest comments and technical correction I would like to see addressed.

Modest comments

- In the introduction the authors mention that global reanalysis with DA, because of horizontal resolution have little capacity for complexities like riverine influences and mesoscale variability. While the authors mention mesoscale variability in the manuscript, I was left wondering if the MOANA hindcast is able to capture riverine influences in ocean dynamics and if authors have tried to analyze this.
- I think in general the methodology section can benefit from a summarizing table or flowchart containing the many datasets (including for the model evaluation datasets) the research is using.
- Line 214-215: Add a sentence or two why and when it is to be expected that higher resolution leads to larger variances (seasonal, peaks, gradient is large).
- Line 240-241: It is stated that patterns of RSME are similar to patterns observed in variance of SST of Fig 3. Whereas there are some similar patterns, notably the highest RMSE (43S-174E and 48S-166E) fall in relatively low variance areas which contradicts the statement. Please elaborate on this.
- Line 259-261: Elaborate how the surface forcing and boundary conditions from GLORYS could trickle down to observed difference in salinity.

Technical correction:

- Line 68 and 70 uses the abbreviation DA, but it is not written in full. Do the authors mean Data Assimilation?
- Line 6 and 7: "sea surface temperature (SST), sea surface height (SSH)" should be with capitals.
- Line 37: add "Due 'to' the large".
- Line 144: the link is not working for me, maybe better to just provide the link to the product store (https://resources.marine.copernicus.eu/products) and give the dataset name along with it instead of this large link.
- Line 177: SST is already in use in the introduction.
- For example, in Figs 2-3-4, what is meant with Moana 'Backbone', please provide context.
- Please provide contour boundaries in the caption of Fig 3.
- Line 239: Further explored in which section?
- Line 259-261: Sentence doesn't flow well.
- Line 285 and 336: Question mark.
- Line 318: sections 1 without the 's'. Also line 354
- Figure 10a: section 8 is missing
- Table 4: Denote separations with A-D in the Table

---

## Author Response (AR1)

**REVIEWER 1**

EGUsphere, referee comment RC1 https://doi.org/10.5194/egusphere-2022-41-RC1, 2022 © Author(s) 2022. This work is distributed under the Creative Commons Attribution 4.0 License. Comment on egusphere-2022-41 Anonymous Referee #1 Referee comment on "Moana Ocean Hindcast – a 25+ years simulation for New Zealand Waters using the ROMS v3.9 model" by Joao Marcos Azevedo Correia de Souza et al., EGUsphere, https://doi.org/10.5194/egusphere-2022-41-RC1, 2022 General comment: The manuscript entitled "Moana Ocean Hindcast – a 25+ years simulation for New Zealand Waters using the ROMS v3.9 model" by Souza et al. presents an extensive model-data comparison of a high-resolution ocean model for the New Zealand region. The comparisons highlight the improvements of the high-resolution simulation over the global data-assimilative model used for boundary conditions. The manuscript is well-structured, moving from the large-scale to the coastal ocean, however it is my opinion that minor changes are needed before this manuscript can be considered for publication. I appreciated the manuscript for being well-structured and for trying to present a substantial amount of information in a succinct way. However, in doing so some of the results aren't explored in any detail and some of the model-data comparisons are glossed over (e.g. SST). The Conclusion sections is a bit disappointing. This section could be improved by a) highlighting the strengths and short-comings of the model and b) by including some of the ways in which the model can be improved on.

**We would like to thank the reviewer for the detailed comments. They were very helpfull to improve the manuscript. We modified the conclusions to make the current simulation improvements clearer.**

Specific comments:

Page 1, Line 20: The authors state that the New Zealand fishing industry is expanding to include coastal aquaculture. New Zealand already has extensive coastal aquaculture activities based around three marine species: salmon, mussels and oysters. However, the Government's Aquaculture Strategy aims to a) expand and improve the existing aquaculture activities, b) advance land-based aquaculture and c) extend aquaculture into the open ocean by 2035. The aquaculture industry in New Zealand are already exploring moving further offshore (so-called open ocean aquaculture) due to coastal warming and the impacts of marine heat waves.

**We modified the text in the introduction to meet the reviewer comment.**

Page 2, Line 46: The statement "revolutionize the understanding and prediction of ocean processes in New Zealand" seems to be a bit of a bold claim. Regional models like the one described here are well-used to understand ocean processes in a range of environments. I agree with the author's underlying view that this simulation will significantly contribute to improve our understanding and predictability of ocean processes but to claim that it will be revolutionary is an exaggeration.

**This is actually the objective of the Moana Project, from which this simulation is a component. This phrase is part of the Project proposal aproved by the funding agency (MBIE). We changed the text to make this clear.**

Page 2, Lines 48-49: What do the authors mean by "7 days forecasts daily"? Were daily forecasts created for a period of 7 days?

**Exactly. Every day we run the model and generate 7 days forecast, that are publicly availabe.**

Page 3, Line 74: Please clarify what you mean by mass structure

**We are referring to the density strucgure – modified in the text.**

Page 6, Line 104: Please be more specific with respect to the local sources that were used for the model bathymetry.

**Added: "such as navigation charts and echo sounder surveys"**

Page 6, Line 107: The authors mention that PGE associated bottom velocities were used to determine which grid cells required bathymetric smoothing. Since this is not a commonly used method in ROMS, please provide more details about this method.

**This is actualy a common approach that is coded in popular libraries that work with the ROMS model, such as pyroms and seapy. We added the reference to 2 papers that use this approach.**

Page 6, Line 118: You use Souza et al. (2020) to justify using CFSR for the atmospheric forcing. However, Souza et al. (2020) only compared the ocean component of CFSR to other ocean models and did not look at the atmospheric component. Is there any evidence that the atmospheric component of CFSR out-competes other atmospheric reanalysis products such as JRA-55 in the New Zealand region?

**Not that we are aware. We removed the reference to Souza (2020) and added a quick explanation for why we use CFSR (well known and lots of use, and long time series).**

Page 10, Lines 208-209: Please be more specific with regards to the location of the high and low SSH centres and the large-scale currents indicated by these SSH centres?

**Added the position of the centres and a short description of the associated circulation features.**

Page 12, Line 230: The authors note that it is interesting that the hindcast reproduced the 2018 temperature peak. This suggest that they didn't expect the model to reproduce this event. Please elaborate why this is particularly noteworthy.

**It is not that we didn't expect, but this gives an insight that this was probably forced by anomalous atmospheric fluxes that were represented by CFSR. A deeper analysis is needed to go beyond expeculation, which is included in a couple of papers focusing on marine heatwaves already submitted for publication.**

Page 12, Line 234-235: The authors mention that the bias pattern is similar to that found for GLORYS in Figure 7 of Souza et al. (2020) but then don't give an explicit description of the biases in the Moana Ocean Hindcast shown in Figure 6.

**A short description was added.**

Page 13, Lines 238-240: While (a) and (b) are the result of intrinsic limitations of the model. Also, the section number for the section addressing (c) is missing. Point (c) is listed as "issues related to the observations of SST from satellites close to the coast and in a region notorious for its high cloud coverage", yet Section 3.4.2. which I suspect is suppose to look at this limitation in more detail only looks at daily sea surface temperature at a number of coastal stations around New Zealand, and doesn't examine the limitations of satellite SST close to the coast.

**The problem with the section number was related to Latex compilation. This was solved now. The idea is to provide a comparison to an independent in situ observational dataset – what is done in section 3.4.2. Both the in situ and satellite are daily mean, what seemed to make more sense from a compatibility point of view and being this a national scale 5km resolution model.**

Page 13, Table 3: RMSE and MAE are standard statistics used in model validation but are often calculated in different ways. Furthermore, maximum-absolute-error is not a commonly used statistic. I recommend that the equations for all the statistics presented in the manuscript are included.

**The standard equations for RMSE and MAE were added o teh manuscript.**

Page 14, Line 251: Was the model output used for the comparison co-located in space and time with the temperature and salinity profiles or was the time-averaged model output extracted at the location of the profiles?

**We co-located by searching for the closest time and linearly interpolating in space.**

Page 14, Line 259-260: The phrasing makes this sentence hard to understand. I think the authors are trying to say something along the lines of "Despite the difficulty in asserting the reasons behind such differences, the differences seem to be in part related to the surface forcing from CFSR and in part due to the boundary conditions from GLORYS.

**Modifyed following the reviewer suggestion.**

Page 17: The section on MLD really gives a nice description of what is observed in the observations and how the model differs from these observations. It would have been really good if this same approach was used throughout the manuscript.

**At the time we decided different approaches would give a more general overview of the simulation performance.**

Page 19, Lines 309-311: It is clear from the statement here that you are cognisant of the fact that volume transport calculation are sensitive to the horizontal and vertical extent over which it is calculated. Was any effort made to try and match the sections used here to the spatial extent over which transport has been calculated in the literature for a better comparison?

**Yes, but we found different methodologies were used accross the literature. Therefore, we decided on an approach that will be used in future publications using this dataset.**

Figure 11: Can you provide any explanation for some of the large differences observed between the observations and the model (e.g. stations 10-15 in subplot G)?

**We don't think these are large differences - please note the different y axis between subplots. The reason for the differences is hard to estimate, but the relatively coarse representation of the coastal geometry and bathymetry in a 5km grid model should play a role.**

Page 27, Line 425-426: "Satellite-derived annual cycles show coastal regions around New Zealand differ from…" This sentence is hard to understand due to the phrasing and requires some clarification.

**Modifyed to improve clarity.**

Page 28, Line 431-432: Here you claim that additional analysis of the coastal SST time- series are needed to understand the latitudinal decrease in SST and this might certainly be the case. However, one contributing factor to the latitudinal decrease would be a change in the background surface water masses as one moves from north to south with warm subtropical water in the north and cooler sub-Antarctic water in the south.

**Thank you for the suggestion.**

Page 28, Line 444: "The model results outperform the global models in the coastal region". This is a very generalised statement that can't really be backed up by any of the results presented in the manuscript. The results presented here only compares the temporal mean SSH and variance of SSH of the Moana Ocean Hindcast to that of one global model, GLORYS.

**Although an improvement can be noticed comparing the present results against the maps presented in Souza et al (2020), we chnaged the text to "The model performs well in the coastal region as demonstrated by the comparison against shore stations."**

The above comments and recommendations along with some technical comments are provided in the attached pdf. Please also note the supplement to this comment: https://egusphere.copernicus.org/preprints/egusphere-2022-41/egusphere-2022-41-RC1-supplement.pdf Powered by TCPDF (www.tcpdf.org)

**REVIEWER 2**

General comment -

Overall this is a good and detailed paper on a hindcast dataset while providing background information and validation of the dataset. The methods undertaken are robust and I commend the authors for their interesting study. As is mentioned in the manuscript, the work builds on earlier work and extensively compares model results with other data sources (like GLORYS). The work presented in novel and provides a major step in the direction of predictive models and I think it is worthy of publishing in NHESS. However, I have listed a couple of modest comments and technical correction I would like to see addressed.

Modest comments –

In the introduction the authors mention that global reanalysis with DA, because of horizontal resolution have little capacity for complexities like riverine influences and mesoscale variability. While the authors mention mesoscale variability in the manuscript, I was left wondering if the MOANA hindcast is able to capture riverine influences in ocean dynamics and if authors have tried to analyze this.

**There are obvious impacts in the Firth of Thames, where horizontal stratification is noticible (and you can see it in the Moana Hindcast). But we had no coastal salinity data to compare to. Our only source was Argo float profiles that don't cover regions shallower than 1000m. This is why choose to not**

- I think in general the methodology section can benefit from a summarizing table or flowchart containing the many datasets (including for the model evaluation datasets) the research is using.

**A summarizing table was added to the text.**

- Line 214-215: Add a sentence or two why and when it is to be expected that higher resolution leads to larger variances (seasonal, peaks, gradient is large).

**We added a short explanation.**

- Line 240-241: It is stated that patterns of RSME are similar to patterns observed in variance of SST of Fig 3. Whereas there are some similar patterns, notably the highest RMSE (43S-174E and 48S-166E) fall in relatively low variance areas which contradicts the statement. Please elaborate on this.

**The 2 regions correspond to fronts of the Southland Current where large SST gradients are present. Therefore, we estimate that the RMSE can be related to difference sin the location of the front in the simulation. Although this can be caused by errors in the model, one should keep in mind that the 1/4 optimal interpolation SST product will have smoothed fronts that will contribute to the large RMSE.**

- Line 259-261: Elaborate how the surface forcing and boundary conditions from GLORYS could trickle down to observed difference in salinity.

**The boundary conditions from GLORYS set the large scale water masses structure that is fed to the model domain. However, the presence of a water mass formation zone in the Subpropical Front provides a pathway through which atmospheric signals coming from**

**CFSR can penetrate to depths bellow the thermocline and influence the 3D density structure - especially for central and deep waters.**

Technical correction:

**All technical corrections were edited in the manuscript.**

- Line 68 and 70 uses the abbreviation DA, but it is not written in full. Do the authors mean Data Assimilation?
- Line 6 and 7: "sea surface temperature (SST), sea surface height (SSH)" should be with capitals.
- Line 37: add "Due 'to' the large".
- Line 144: the link is not working for me, maybe better to just provide the link to the product store (https://resources.marine.copernicus.eu/products) and give the dataset name along with it instead of this large link.
- Line 177: SST is already in use in the introduction. - For example, in Figs 2-3-4, what is meant with Moana 'Backbone', please provide context.

**It is another name for the Moana Ocean Hindcast. A reference was provided on its first apearance in the text.**

- Please provide contour boundaries in the caption of Fig 3.
- Line 239: Further explored in which section?
- Line 259-261: Sentence doesn't flow well.
- Line 285 and 336: Question mark.
- Line 318: sections 1 without the 's'. Also line 354
- Figure 10a: section 8 is missing
- Table 4: Denote separations with A-D in the Table

---

## Referee Report (RR1)

**General comment:**
I commend the authors for adequately addressing the major comments from the first round of reviews. However, it was a bit disappointing to note that none of the technical comments in the pdf were addressed. The amendments done thus far has definitely improved the manuscript, putting it one step closer to publication. It is my opinion that some minor changes and technical corrections are required before this manuscript can be considered for publication.

**Specific comments:**

Page 2, Line 46: The statement "revolutionize the understanding and prediction of ocean processes in New Zealand" seems to be a bit of a bold claim. Regional models like the one described here are well-used to understand ocean processes in a range of environments. I agree with the author's underlying view that this simulation will significantly contribute to improve our understanding and predictability of ocean processes but the claim that it will be revolutionary at this stage lacks supportive evidence.

Page 2, Lines 48-49: Rephrase "7 days forecasts daily" to make it more obvious that daily forecasts are created for a period of 7 days.

Page 18, Lines 328-330: The phrasing of this sentence makes it very hard to understand. The description in Table 5's caption makes much more sense. This sentence should be rephrased so that it is easier to understand how the transport was calculated along the sections.

Figure 10: Even though it is stated in the caption that section 8 can not be displayed on the Figure, I think more effort should be put towards indicating where the section is located. Without this information the reader has no idea of where the Cook Strait section is located.

Page 22, Line 360: The model volume transport calculated for the ECC south section is significantly higher than that reported by Fernandez et al. (2018). How does the model volume transport compare to that of Fernandez et al. (2018) if calculated along similar sections as those used by Fernandez et al. (2018)?

**Technical corrections:**

Page 2, Line 25-26: Move the second set of references (e.g. Chaput et al. 2022; Silva et al., 2019) to the end of the sentence.

Page 2, Line 29: **an** important source

Page 2, Lines 48-49: I suggest changing 'please visit the project website for details' to 'details available on the project website'

Page 2, Line 55: replace bibliography with literature

Page 3, Line 58: Change Marine Heat Wave to marine heatwave as used throughout the rest of the manuscript.

Page 3, Line 59: Remove the quotation marks around "sea surface temperature"

Page 3, Ling 60: **n**orth of the Subtropical Front

Page 3, Line 61: Please provide the name of this western boundary current

Page 3, Lines 75-78: Please provide references to substantiate these statements.

Page 3, Line 76: **Three K**ings Islands

Page 3, Line 76: Change mass structure to density structure

Page 3, Line 80: Change following publications to subsequent publications or future publications.

Page 4, Line 90: Replace semi-colon between references with a comma

Page 4, Line 94: Unbold 5km resolution

Page 4, Line 96: Chatham **I**slands

Page 4, Line 96: **three** New Zealand main islands. Single-digit numbers should preferably be spelled out.

Page 4, Line 97: **western** boundary

Figure 1: **Puysegur**

Page 6, Line 103: Unbold 50 vertical layers. Also, ideally a sentence should not start with numerical digits but if it does then the number should be written out.

Page 6, Line 1227 **(**Lellouche et al. 2021**)**

Page 6, Line 128: 1 day$^{-1}$

Page 6, Line 128: Change warm-up period to spin-up period. Did the model reach a steady state after just 1 year's integration? I suspect that it did but it did.

Page 7, Line 137: **four** readily available global ocean **reanalyses**

Page 7, Line 139: **four** of the

Page 7, Line 145: SST was defined in the introduction so there is no need to do it again here.

Page 7, Line 149: **the** Mercator Ocean

Page 7, Line 150: can be determinant **factors** for the representation of the  sea surface height

Page 7, Line 155: representation of the

Page 7, Line 165: The Moana Ocean Hindcast

Page 8, Line 175: **T**able 1, and a description **of each is** provided in **the** next subsections

Page 8, Line 184: SST has already been defined thus no need to define it again.

Page 8, Line 191: Change from to including

Page 8, Line 192: Replace the semi-colons with commas.

Page 8, Line 193: Change forward to onward

Page 9, Line 202: collected by  New Zealand's

Page 10, Line 209: (NZ-OOS; **O**'Callaghan et al., 2019)

Page 10, Line 217: Add the acronym for the East Auckland Current here.

Page 10, Line 220: **es**pecially in the region

Page 10, Line 220: If the acronym EAUC is included in line 217 as suggested then you can just use the acronym here.

Page 11, Lines 226-227: Combine these two sentences: leading to stronger variability**,** **a**nd less is left

Page 11, Line 234: Unbold 40 days

Page 11, Line 235: altimetry maps **at** the equator

Page 11, Line 236: days **around** New Zealand

Page 11, Line 238: pattern is  similar

Page 11, Line 241: **inter-annual** variability

Page 11, Line 242: variability **at**

Figure 3, Caption: from the free-**r**unning Moana Ocean Hindcast (left)**,** the data-assimilating

Figure 4: In Figure 3, the first panel was the Moana Backbone followed by GLORYS and then AVISO.  It would make it easier to follow along if the order is kept the same between figures.

Figure 4 caption: G**L**ORYS

Page 13, Line 244: events with duration **on** the order

Page 13, Line 253: While (a) and (b) **are**

Page 13, Line 256: These **two** regions

Page 13, Line 261: summarized in Table **4**

Page 13, Lines 262: Please rephrase this sentence so that it is clearer that SSH errors are similar to that from global simulations, while SST performs better. It will also help if the statistics for GLORYS is included in Table 4.

Figure 6, Caption: These relate to the fact **that**

Page 15, Lines 269-273: This seems a bit out of place here. I would suggest adding this information in a more succinct way (e.g. only the formulas) into the caption of Table 4.

Page 15, Line 281: with values **in** general**ly** under 1°C

Page 15, Line 282: **below** the mixed layer. These compare well **to** with

Page 15, Line 284: As shown in the RMSE profile **(Figure 7)**

Page 15, Line 288: large scale water **mass** structure

Figure 7 caption: Moana Ocean Hindcast **simulation** in relation to... A zoom **in** of the

Page 16, Line 302: Indicate the location of the plateau and other features mentioned in the text on Figure 1.

Figure 8: At first glance it is a bit confusing to have the depth-ranges used to generate the subplots as the label on the y-axis when in fact the y-axis represents latitude.

Figure 8: Caption: The differences are divided **into** slabs… deep waters (1000-2000m – **bottom** row). A geographic distribution pattern is evident in the model **result** differences

Page 18, Line 310: model**'s**

Page 18, Line 314:  **(**Elzahaby et al. 2021)

Page 18, Line 317: Chiswell et al. (2015) **and** Stevens et al. (2019)

Page 18, Lines 336: **comparisons** with estimates presented in the literature. We limit **the** model assessment

Page 18, Line 336: Change relative to corresponding

Page 18, Line 337: carried **out** to date along the eastern margin of **the** NZ

Table 5: The offshore sections need to be labeled A-D to match the sections in Figure 10 and section 4 should be changed to **W**est coast of North Island. Also, the Cook Strait section is not indicated on Figure 10.
Page 21, Line 344: Change Following to Below

Page 21, Line 345: "remote based" I suggest changing it so that it is clearer that it is volume transport calculated from remotely sensed data.

Page 21, Line 347: Include a cross-reference to Table 5 for the transports reported here.

Page 21, Line 351: using **a** significantly

Page 21, Line 353: along **the** same

Page 21, Line 353-354: This sentence is hard to comprehend. I suggest rephrasing it to something like "These values are also consistent with a volume transport of 8-15 Sv derived from Argo float trajectories in the same region (Bowen et al., 2014)."

Page 21, Line 357: Add a cross-reference to Table 5 for the transports reported here.

Page 22, Line 375: same location as **for** the section

Page 22, Line 376: CTD **surveys**

Page 22, Line 377: carried **out** between  1993 and 2000

Page 22, Line 382: cross-sectional transport **through** the Cook Strait

Page 22, Line 384: Include a cross-reference to Table 4 for the transports reported here.

Page 23, Line 395: Everywhere else throughout the manuscript it is referred to as the Moana Ocean Hindcast. I suggest changing ROMS hindcast to Moana Ocean Hindcast to keep it consistent throughout the manuscript.

Page 23, Line 397: Results from **the** harmonic analysis

Page 23, Line 398: constituent**;** Figure 11)

Figure 11: The black and red markers are different sizes with the red markers larger than the black markers. Is there a reason for this? If not, then I suggest making the markers the same size.

Page 24, Line 407: Root-Mean-Square-Error (RMSE). Abbreviations should be written out at the first instance where they are used and not towards the end of the manuscript.

Page 24, Line 412: The RMSE of **the** phase

Page 24, Line 413: The **phase of the** semidiurnal (diurnal) constituent , $K_2$ ($P_1$) had an RMSE  of

Page 25, Line 416: indicator of  oceanic processes **such** as
Page 25, Line 420: Everywhere else throughout the manuscript it is referred to as the Moana Ocean Hindcast. I suggest changing ROMS hindcast to Moana Ocean Hindcast to keep it consistent with your vocabulary.

Page 25, Line 420: time series from **three** locations

Page 25, Line 426: denoted **by** MSE

Page 25, Line 429: , North Cape ($SLA_{TG12}$, not shown)**,**

Page 26, Line 432: Was the modelled data extracted at the grid cell closest to the station locations? If so, this should be included in the manuscript for both the temperature and sea level stations.

Page 26, Line 439: Portobello**;** Figure 13 G, I)

Page 28, Line 451:452: I suggest rephrasing this sentence to make it clearer that the model under-estimate the seasonal cycle (i.e. cooler (warmer) temperatures in summer (winter) than observed) at these stations.

Page 28, Line 454-455: I suggest rephrasing the second part of this sentence e.g. is potentially unresolved in a regional-scale oceanic model of this resolution due to land-air-sea processes.

Table 7 caption: I recommend splitting the last sentence of the caption in two.

Page 28 (0), Line 468: unbold Moana Ocean Hindcast

Page 28 (0), Line 472: series of **analyses**

Page 28 (0), Line 473: Replace shore with coastal temperature and tidal

Page 29 (1), Line 482: **calibrated** for the New Zealand region

Page 29 (1), Line 482-483: I suggest rewriting this sentence to say that it could lead to improvements of the model solution on the continental shelf.

Page 29 (1), Line 485: Remove the dash at the start of the sentence.

---

## Author Response (AR2)

**General                                                                                      comment:**

**I commend the authors for adequately addressing the major comments from the first round of reviews. However, it was a bit disappointing to note that none of the technical comments in the pdf were addressed. The amendments done thus far has definitely improved the manuscript, putting it one step closer to publication. It is my opinion that some minor changes and technical corrections are required before this manuscript can be considered for publication.**

We would like to thank the reviewer for the helpful and detailed comments. We did our best to take all of them into consideration and were able to respond to most of them. We sincerely hope this brings our manuscript to a state where it can be published by GMD.

**Specific                                                                                       comments:**

**Page 2, Line 46: The statement "revolutionize the understanding and prediction of ocean processes in New Zealand" seems to be a bit of a bold claim. Regional models like the one described here are well-used to understand ocean processes in a range of environments. I agree with the author's underlying view that this simulation will significantly contribute to improve our understanding and predictability of ocean processes but the claim that it will be revolutionary at this stage lacks supportive evidence.**

We agree with the core of the reviewer comment. The claim to "revolutionize" was simply a quote from the project proposal. That said, the text was modified to estate we aim to improve the how we understand        and        predict        the        ocean        around        New        Zealand.

**Page 2, Lines 48-49: Rephrase "7 days forecasts daily" to make it more obvious that daily forecasts are created for a period of 7 days.**

Modified following the reviewer's comment.

**Page 18, Lines 328-330: The phrasing of this sentence makes it very hard to understand. The description in Table 5's caption makes much more sense. This sentence should be rephrased so that it is easier to understand how the transport was calculated along the sections.**

This text has been clarified.

**Figure 10: Even though it is stated in the caption that section 8 can not be displayed on the Figure, I think more effort should be put towards indicating where the section is located. Without this information the reader has no idea of where the Cook Strait section is located.**

The section was added to the figure

**Page 22, Line 360: The model volume transport calculated for the ECC south section is significantly higher than that reported by Fernandez et al. (2018). How does the model volume transport compare to that of Fernandez et al. (2018) if calculated along similar sections as those used by Fernandez et al. (2018)?**

A detailed discussion of the differences in calculation methods and the significantly higher transport estimates compared to previous estimates in the literature, including Fernandez et al. 2018, is presented in Kerry et al. 2022. This reference has been included in the text. Here we explain how our estimates of ECC transport are considerably greater than estimates presented to date in the literature where key differences in the calculation methods and locations exist. Specifically, these attempts to

estimate transport use satellite altimetry combined with subsurface observations to estimate the vertical structure of the current, and assume a level of no motion of 2000dbar. However, we show that the ECC extends below 2000m, consistent with other studies that have found substantial velocities below 2000 m in the EAUC and ECC regions. Furthermore, previous estimates have attempted to separate the recirculation driven by the semi-permanent eddies, while our estimates include this recirculation, and transport estimates are highly sensitive to the distance offshore over which transport is calculated. We direct the reviewers to this reference for a detailed discussion with references.

**Technical                                                                                         corrections:**

**Page 2, Line 25-26: Move the second set of references (e.g. Chaput et al. 2022; Silva et al., 2019) to the end of the sentence.**

Modified following the reviewer's comment.

**Page 2, Line 29: an important source**

Modified following the reviewer's comment.

**Page 2, Lines 48-49: I suggest changing 'please visit the project website for details' to 'details available on the project website'**

Modified following the reviewer's comment.

**Page 2, Line 55: replace bibliography with literature**

Modified following the reviewer's comment.

**Page 3, Line 58: Change Marine Heat Wave to marine heatwave as used throughout the rest of the manuscript.**

Modified following the reviewer's comment.

**Page 3, Line 59: Remove the quotation marks around "sea surface temperature"**

Modified following the reviewer's comment.

**Page 3, Ling 60: north of the Subtropical Front**

Modified following the reviewer's comment.

**Page 3, Line 61: Please provide the name of this western boundary current**

East Auckland Current – added to the text

**Page 3, Lines 75-78: Please provide references to substantiate these statements.**

References were added to each statement.

**Page 3, Line 76: Three Kings Islands**

Modified following the reviewer's comment.

**Page 3, Line 76: Change mass structure to density structure**

Modified following the reviewer's comment.

**Page 3, Line 80: Change following publications to subsequent publications or future publications.**

Modified following the reviewer's comment.

**Page 4, Line 90: Replace semi-colon between references with a comma**

Modified following the reviewer's comment.

**Page 4, Line 94: Unbold 5km resolution**

Modified following the reviewer's comment.

**Page 4, Line 96: Chatham Islands**

Modified following the reviewer's comment.

**Page 4, Line 96: three New Zealand main islands. Single-digit numbers should preferably be spelled out.**

Modified following the reviewer's comment.

**Page 4, Line 97: western boundary**

Modified following the reviewer's comment.

**Figure 1: Puysegur**

Modified following the reviewer's comment.

**Page 6, Line 103: Unbold 50 vertical layers. Also, ideally a sentence should not start with numerical digits but if it does then the number should be written out.**

Modified following the reviewer's comment.

**Page 6, Line 1227 (Lellouche et al. 2021)**

Modified following the reviewer's comment.

**Page 6, Line 128: 1 day-1**

Modified following the reviewer's comment.

**Page 6, Line 128: Change warm-up period to spin-up period. Did the model reach a steady state after just 1 year's integration? I suspect that it did but it did.**

Modified following the reviewer's comment. Yes, the model was actualy stable after less than that. But we chose 1 year to be conservative.

**Page 7, Line 137: four readily available global ocean reanalyses**

Modified following the reviewer's comment.

**Page 7, Line 139: four of the**

Modified following the reviewer's comment.

**Page 7, Line 145: SST was defined in the introduction so there is no need to do it again here.**

Modified following the reviewer's comment.

**Page 7, Line 149: the Mercator Ocean**

Modified following the reviewer's comment.

**Page 7, Line 150: can be determinant factors for the representation of the the sea surface height**

Modified following the reviewer's comment.

**Page 7, Line 155: representation of the the**

Modified following the reviewer's comment.

**Page 7, Line 165: The Moana Ocean Hindcast hindcast**

Modified following the reviewer's comment.

**Page 8, Line 175: Table 1, and a description of each is provided in the next subsections**

Modified following the reviewer's comment.

**Page 8, Line 184: SST has already been defined thus no need to define it again.**

Modified following the reviewer's comment.

**Page 8, Line 191: Change from to including**

Modified following the reviewer's comment.

**Page 8, Line 192: Replace the semi-colons with commas.**

Modified following the reviewer's comment.

**Page 8, Line 193: Change forward to onward**

Modified following the reviewer's comment.

**Page 9, Line 202: collected by the New Zealand's**

Modified following the reviewer's comment.

**Page 10, Line 209: (NZ-OOS; O'Callaghan et al., 2019)**

Modified following the reviewer's comment.

**Page 10, Line 217: Add the acronym for the East Auckland Current here.**

Modified following the reviewer's comment.

**Page 10, Line 220: especially in the region**

Modified following the reviewer's comment.

**Page 10, Line 220: If the acronym EAUC is included in line 217 as suggested then you can just use the acronym here.**

Modified following the reviewer's comment.

**Page 11, Lines 226-227: Combine these two sentences: leading to stronger variability, and less is left**

Modified following the reviewer's comment.

**Page 11, Line 234: Unbold 40 days**

Modified following the reviewer's comment.

**Page 11, Line 235: altimetry maps at the equator**

Modified following the reviewer's comment.

**Page 11, Line 236: days around New Zealand**

Modified following the reviewer's comment.

**Page 11, Line 238: pattern is the similar**

Modified following the reviewer's comment.

**Page 11, Line 241: inter-annual variability**

Modified following the reviewer's comment.

**Page 11, Line 242: variability at Figure 3, Caption: from the free-running Moana Ocean Hindcast (left), the data-assimilating**

Modified following the reviewer's comment. Also added parenthesis for the GLORYS and AVISO.

**Figure 4: In Figure 3, the first panel was the Moana Backbone followed by GLORYS and then AVISO. It would make it easier to follow along if the order is kept the same between figures.**

We were unable to change the order of the subplots, since this would require re-processing a large amount of data to re-generate the figures. Since this does not compromise the text understanding, we opted to keep the original figure.

**Figure 4 caption: GLORYS**

Sorry, but I did not understand the reviewer suggestion.

**Page 13, Line 244: events with duration on the order**

Modified following the reviewer's comment.

**Page 13, Line 253: While (a) and (b) are**

Modified following the reviewer's comment.

**Page 13, Line 256: These two regions**

Modified following the reviewer's comment.

**Page 13, Line 261: summarized in Table 4**

Modified following the reviewer's comment.

**Page 13, Lines 262: Please rephrase this sentence so that it is clearer that SSH errors are similar to that from global simulations, while SST performs better. It will also help if the statistics for GLORYS is included in Table 4.**

We modified the sentence for clarity. We chose not to add the GLORYS statistics, since these are already included in a previous publication. The idea here is to focus on the evaluation of the Moana Hindcast against observations as much as possible.

**Figure 6, Caption: These relate to the fact that**

Modified following the reviewer's comment.

**Page 15, Lines 269-273: This seems a bit out of place here. I would suggest adding this information in a more succinct way (e.g. only the formulas) into the caption of Table 4.**

I tryied to add the equations to the caption, but it looked too cramped up and without enough space to explaing the variables in the equations. Therefore, I ended up opting to keep the original text.

**Page 15, Line 281: with values in generally under 1ºC**

Modified following the reviewer's comment.

**Page 15, Line 282: below the mixed layer. These compare well to with**

Modified following the reviewer's comment.

**Page 15, Line 284: As shown in the RMSE profile (Figure 7)**

Modified following the reviewer's comment.

**Page 15, Line 288: large scale water mass structure**

Modified following the reviewer's comment.

**Figure 7 caption: Moana Ocean Hindcast simulation in relation to... A zoom in of the**

Modified following the reviewer's comment.

**Page 16, Line 302: Indicate the location of the plateau and other features mentioned in the text on Figure 1.**

Locations were added to the map in Figure 1.

**Figure 8: At first glance it is a bit confusing to have the depth-ranges used to generate the subplots as the label on the y-axis when in fact the y-axis represents latitude.**

This figure follows the same logic as in previous publication looking at global simulations performance. Therefore, for comparison's sake we decided to not change it.

**Figure 8: Caption: The differences are divided into slabs... deep waters (1000-2000m – bottom row). A geographic distribution pattern is evident in the model result differences**

Modified following the reviewer's comment.

**Page 18, Line 310: model's**

Modified following the reviewer's comment.

**Page 18, Line 314: (Elzahaby et al. 2021)**

Modified following the reviewer's comment.

**Page 18, Line 317: Chiswell et al. (2015) and Stevens et al. (2019)**

Sorry, but couldn't figure how to do this in LaTex.

**Page 18, Lines 336: comparisons with estimates presented in the literature. We limit the model assessment**

Modified following the reviewer's comment.

**Page 18, Line 336: Change relative to corresponding**

Modified following the reviewer's comment.

**Page 18, Line 337: carried out to date along the eastern margin of the NZ**

Modified following the reviewer's comment.

**Table 5: The offshore sections need to be labeled A-D to match the sections in Figure 10 and section 4 should be changed to West coast of North Island. Also, the Cook Strait section is not indicated on Figure 10.**

The table was modified accordingly and the Cook Strait section was added to Figure 10.

**Page 21, Line 344: Change Following to Below**

Modified following the reviewer's comment.

**Page 21, Line 345: "remote based" I suggest changing it so that it is clearer that it is volume transport calculated from remotely sensed data.**

Modified following the reviewer's comment.

**Page 21, Line 347: Include a cross-reference to Table 5 for the transports reported here.**

Reference added.

**Page 21, Line 351: using a significantly**

Modified following the reviewer's comment.

**Page 21, Line 353: along the same**

Modified following the reviewer's comment.

**Page 21, Line 353-354: This sentence is hard to comprehend. I suggest rephrasing it to something like "These values are also consistent with a volume transport of 8-15 Sv derived from Argo float trajectories in the same region (Bowen et al., 2014)."**

Modified following the reviewer's comment.

**Page 21, Line 357: Add a cross-reference to Table 5 for the transports reported here.**

Reference added.

**Page 22, Line 375: same location as for the section**

Modified following the reviewer's comment.

**Page 22, Line 376: CTD surveys**

Modified following the reviewer's comment.

**Page 22, Line 377: carried out between years 1993 and 2000**

Modified following the reviewer's comment.

**Page 22, Line 382: cross-sectional transports through the Cook Strait**

Modified following the reviewer's comment.

**Page 22, Line 384: Include a cross-reference to Table 4 for the transports reported here.**

Reference added.

**Page 23, Line 395: Everywhere else throughout the manuscript it is referred to as the Moana Ocean Hindcast. I suggest changing ROMS hindcast to Moana Ocean Hindcast to keep it consistent throughout the manuscript.**

Modified following the reviewer's comment.

**Page 23, Line 397: Results from the harmonic analysis**

Modified following the reviewer's comment.

**Page 23, Line 398: constituent;**

Modified following the reviewer's comment.

**Figure 11) Figure 11: The black and red markers are different sizes with the red markers larger than the black markers. Is there a reason for this? If not, then I suggest making the markers the same size.**

The markers have different sizes to make them visible when overlaping.

**Page 24, Line 407: Root-Mean-Square-Error (RMSE). Abbreviations should be written out at the first instance where they are used and not towards the end of the manuscript.**

Modified following the reviewer's comment.

**Page 24, Line 412: The RMSE of the phase error**

Modified following the reviewer's comment.

**Page 24, Line 413: The phase of the semidiurnal (diurnal) constituent error, K2 (P1) had an RMSE phase of**

Modified following the reviewer's comment.

**Page 25, Line 416: indicator of such oceanic processes such as**

Modified following the reviewer's comment.

**Page 25, Line 420: Everywhere else throughout the manuscript it is referred to as the Moana Ocean Hindcast. I suggest changing ROMS hindcast to Moana Ocean Hindcast to keep it consistent with your vocabulary.**

Modified following the reviewer's comment.

**Page 25, Line 420: time series from three locations**

Modified following the reviewer's comment.

**Page 25, Line 426: denoted by MSE**

Modified following the reviewer's comment.

**Page 25, Line 429: for example, North Cape (SLATG12, not shown),**

Modified following the reviewer's comment.

**Page 26, Line 432: Was the modelled data extracted at the grid cell closest to the station locations? If so, this should be included in the manuscript for both the temperature and sea level stations.**

Yes, the simulation results were extracted from the closest valid (water) grid point. This was included in the manuscript for both the SSH and Temperature comparisons.

**Page 26, Line 439: Portobello; Figure 13 G, I)**

Modified following the reviewer's comment.

**Page 28, Line 451:452: I suggest rephrasing this sentence to make it clearer that the model underestimate the seasonal cycle (i.e. cooler (warmer) temperatures in summer (winter) than observed) at these stations.**

Simply stating that there are differences in the temperature doesn't transmit the information that these are only bout 0.25C. therefore, we decided to keep the quantitative differences and added a sentence reflecting the reviewer comment.

**Page 28, Line 454-455: I suggest rephrasing the second part of this sentence e.g. is potentially unresolved in a regional-scale oceanic model of this resolution due to land-air-sea processes.**

Modified following the reviewer's comment.

**Table 7 caption: I recommend splitting the last sentence of the caption in two.**

Modified following the reviewer's comment.

**Page 28 (0), Line 468: unbold Moana Ocean Hindcast**

Modified following the reviewer's comment.

**Page 28 (0), Line 472: series of analyses**

Modified following the reviewer's comment.

**Page 28 (0), Line 473: Replace shore with coastal temperature and tidal**

Modified following the reviewer's comment.

**Page 29 (1), Line 482: calibrated for the New Zealand region**

Modified following the reviewer's comment.

**Page 29 (1), Line 482-483: I suggest rewriting this sentence to say that it could lead to improvements of the model solution on the continental shelf.**

Rephrased.

**Page 29 (1), Line 485: Remove the dash at the start of the sentence.**

Removed.